# Precision therapeutic targeting of human cancer cell motility

Li Xu[1,11], Ryan Gordon[2], Rebecca Farmer[3], Abhinandan Pattanayak[2], Andrew Binkowski[4], Xiaoke Huang[1], Michael Avram[5], Sankar Krishna[1], Eric Voll[1], Janet Pavese[1], Juan Chavez[6], James Bruce[6], Andrew Mazar[3], Antoinette Nibbs[3], Wayne Anderson[7], Lin Li[8], Borko Jovanovic[9], Sean Pruell[1], Matias Valsecchi[1], Giulio Francia[10], Rick Betori[3], Karl Scheidt [3] & Raymond Bergan[2]

Increased cancer cell motility constitutes a root cause of end organ destruction and mortality, but its complex regulation represents a barrier to precision targeting. We use the unique characteristics of small molecules to probe and selectively modulate cell motility. By coupling efficient chemical synthesis routes to multiple upfront in parallel phenotypic screens, we identify that KBU2046 inhibits cell motility and cell invasion in vitro. Across three different murine models of human prostate and breast cancer, KBU2046 inhibits metastasis, decreases bone destruction, and prolongs survival at nanomolar blood concentrations after oral administration. Comprehensive molecular, cellular and systemic-level assays all support a high level of selectivity. KBU2046 binds chaperone heterocomplexes, selectively alters binding of client proteins that regulate motility, and lacks all the hallmarks of classical chaperone inhibitors, including toxicity. We identify a unique cell motility regulatory mechanism and synthesize a targeted therapeutic, providing a platform to pursue studies in humans.

[1] Department of Medicine, Northwestern University, Chicago, IL 60611, USA. [2] Division of Hematology/Oncology, Knight Cancer Institute, Oregon Health & Science University, Portland, OR 97239, USA. [3] Department of Chemistry, Northwestern University, Evanston, IL 60208, USA. [4] Department of Computer Science, University of Chicago, Chicago, IL 60637, USA. [5] Department of Anesthesiology, Northwestern University, Chicago, IL 60611, USA. [6] Department of Genome Sciences, University of Washington, Seattle, WA 98195, USA. [7] Department of Molecular Pharmacology and Biological Chemistry, Northwestern University, Chicago, IL 60611, USA. [8] Department of Pathology, Northwestern University, Chicago, IL 60611, USA. [9] Department of Preventive Medicine, Northwestern University, Chicago, IL 60611, USA. [10] Border Biomedical Research Center, University of Texas at El Paso, El Paso, TX 79968, USA. [11]Present address: Department of Gastroenterology, Xiang'an Hospital of Xiamen University, Fujian 361101 Xiamen, China. These authors contributed equally: Li Xu, Ryan Gordon, Rebecca Farmer. Correspondence and requests for materials should be addressed to R.B. (email: bergan@ohsu.edu)

ncreased cell motility is a fundamental characteristic of cancer cells[1]. It is required in order for cells to invade through the basement membrane, represents an initial step in the metastatic cascade, and is necessary for cells to move from their primary organ of origin to distant metastatic sites. The movement of cancer cells out of their primary organ of origin greatly reduces the chances of survival[2]. Movement of cells to distant organs, and their resultant destruction, constitutes a primary cause of cancer-associated morbidity and mortality[3]. Processes that drive the development of increased cell motility represent high-value therapeutic targets. However, comprehensive endeavors aimed at selectively inhibiting cancer cell motility and resultant metastasis have met with failure[4,5]. While many pathways have been shown to regulate cell motility, they constitute pathways whose regulatory effects are pleiotropic[5]. It has therefore not been possible to identify regulators of cell motility possessing the selective capacity to support targeted manipulation.

Recognizing the critical importance and intractable nature of this problem, we reasoned that it needed to be approached in a unique manner. We did so by considering that small chemicals have potent biological properties, that single atom changes in their structure can affect those properties, that chemical structure can be modulated, and that as such they constitute highly refined biological probes. We hypothesized that we could use them to identify novel and selective sites that regulate cancer cell motility and that such sites would constitute high-value therapeutic targets.

Herein, we delineate a novel and selective regulatory mechanism for these processes using efficient synthesis routes and resultant small chemicals as biological probes. We go on to demonstrate the therapeutic potential of the resultant probe, KBU2046. We do so by demonstrating selectivity across comprehensive molecular, cellular, and systemic assays. Efficacy of KBU2046 is demonstrated across several different in vitro models and across multiple murine models of human cancer metastasis, which includes decreased metastasis, decreased bone destruction, and prolonged survival. Also, comprehensive pharmacokinetic and toxicity studies further support therapeutic potential. Finally, we go on to characterize the molecular mechanism and its ability to perturb the novel regulatory process.

## Results

**Identifying a selective inhibitor of cell motility**. We selected flavonoids as a chemical scaffold to advance probe synthesis because they exert a wide range of biological effects[6]. We began with 4′,5,7-trihydroxyisoflavone (genistein) as our starting point because of its known anti-motility properties. We previously demonstrated that nanomolar concentrations of genistein inhibit human prostate cancer (PCa) cell invasion in vitro[7], metastasis in a murine orthotopic model[8], and in the context of a prospective human trial that it downregulates matrix metalloproteinase 2 (MMP-2) expression in prostate tissue[9]. While its diverse spectrum of biological effects render it unusable as a selective and potent biological probe, these same properties maximize its potential to selectively probe a wide spectrum of bioactive sites upon chemical diversification.

We developed a series of related molecular probes through phenotypically driven structure activity relationship studies, specifically through chemical modification of the genistein structure (aromatic substitution and ring saturation). These compounds were advanced by iterative selection for inhibition of human PCa cell invasion (Fig. 1a and Supplementary Note 1 and Supplementary Fig. 1). A major parallel goal was deselection for inhibition of cell growth (an indicator of off-target effects). Knowing that genistein has estrogenic action, and guided by the

crystal structure of genistein bound to estrogen receptor (ER)[10], we also de-selected for ER-binding. Through this strategy, (±)-3(4-fluorophenyl)chroman-4-one (KBU2046), a halogen-substituted isoflavanone, was discovered (Fig. 1a).

KBU2046 inhibits cell invasion with efficacy equal-to-or-greater-than that of genistein for human prostate cells, including normal prostate epithelial cells, as well as primary and metastatic PCa cells (Fig. 1b). Cell migration is a major determinant of cell invasion[11], and KBU2046 inhibited the migration of human prostate, breast, colon, and lung cancer cells (Fig. 1c). Importantly, KBU2046 had high selectivity in cellular assays. It was not toxic to human prostate cells (Table 1), to human bone marrow stem cells (Fig. 1d), nor to cells in the NCI-60 cell line panel (Supplementary Fig. 2). Bone marrow toxicity is induced by a wide spectrum of therapeutic agents, and is frequently a dose-limiting toxicity for anti-cancer agents. Furthermore, in estrogen-responsive human breast cancer MCF-7 cells, KBU2046 did not activate estrogen-responsive genes (Fig. 1e and Supplementary Fig. 3).

**KBU2046 inhibits metastasis and prolongs survival**. Because metastasis is a systemic process, effective small molecule probes must operate at the systemic level. We designed the probe to contain chemical properties known to be associated with systemically active small molecules (Supplementary Fig. 4). Employing a murine orthotopic implantation model of human PCa previously characterized by us[8], KBU2046 was shown to significantly decrease metastasis in a dose-dependent manner by up to 92%, at plasma concentrations of 1.1–24 nM (Fig. 2a, b). Comprehensive characterization of KBU2046 pharmacokinetics demonstrated maintenance of plasma concentrations >24 nM for 9.3 h after a single oral dose, and allowed for characterization of pharmacokinetic parameters (Fig. 2c and Supplementary Fig. 5). At the systemic level, KBU2046 was a highly selective inhibitor of metastasis. Comprehensive analysis of primary tumor growth, animal behavior, weight, histologic examination of multiple organs, and serum chemistry profiling failed to identify KBU2046-associated off-target effects (Supplementary Figs. 6, 7).

Recognizing the established link between metastasis and decreased survival in humans, we evaluated KBU2046's impact on survival. The orthotopic PCa model exhibits tumor growth around the urogenital tract, inhibiting renal function and precluding assessment of the impact of metastatic burden on survival. However, orthotopic implantation of human breast cancer cells, followed by surgical removal of the resultant primary tumor, provides a murine model wherein survival is dictated by metastatic burden[12]. KBU2046 significantly prolonged the survival of mice treated in a post-surgery adjuvant setting (Fig. 2d).

If KBU2046 were inhibiting metastasis through inhibition of cell motility, then it should have little to no effect upon the metastatic process once cells have implanted into distant organs. Recognizing that skeletal metastases are a major clinical problem with PCa, further assessment of this paradigm was pursued with an established PCa bone metastasis model[13]. PC3 luciferase tagged (PC3-luc) cells were delivered by ultrasound-guided intracardiac (IC) injection and metastatic outgrowth monitored weekly via IVIS imaging (Fig. 3a and Supplementary Fig. 8). Compared to control mice, the pre-cohort of mice (KBU2046 treatment starts 3 days prior to IC injection and continuing through the end of the experiment) experienced a significant decrease in total metastatic burden, as well as decreased metastasis to the mandible (for which this model is designed) (Fig. 3b, c). In contrast, with the post-cohort of mice, metastasis to the total body as well as to the jaw do not differ from control

mice. With the post-cohort of mice, cells are given 3 days post IC injection to invade into distant organ sites before treatment is begun, with treatment then continuing through the end of the experiment. In the Pre7Stop cohort of mice, treatment starts 3 days prior to IC injection, continues through day 7 post IC injection and is not given for the remaining 3 weeks of the

experiment. Findings in this cohort of mice suggest an intermediate outcome between that of pre- and post-cohorts. Specifically, total body metastasis is significantly decreased in the Pre7Stop cohort, compared to both control and post-cohorts. While jaw metastasis is significantly decreased compared to control, it is not significantly decreased compared to the post-

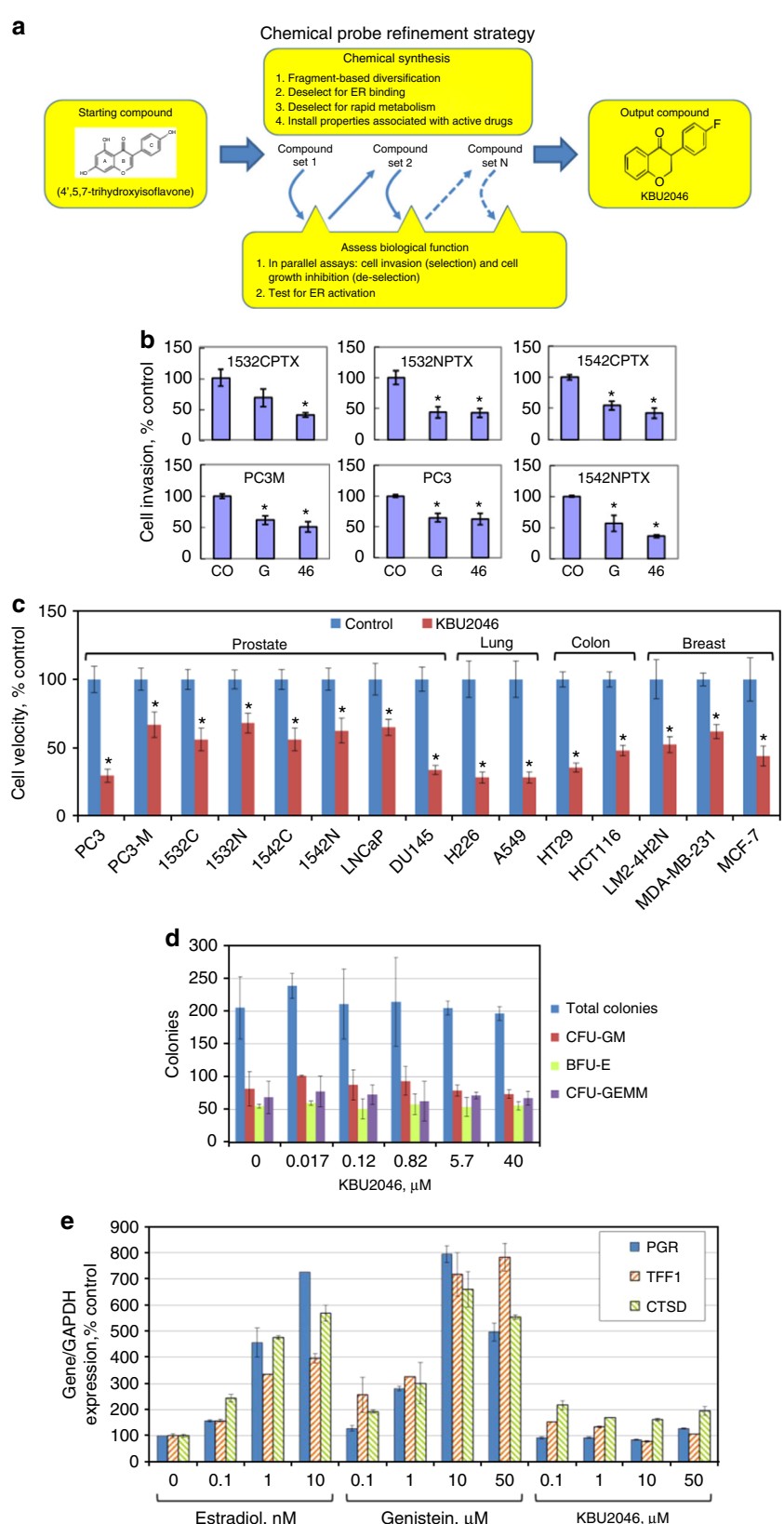

cohort of mice, yet the average value is below that of post-mice and is approaching that of the pre-cohort of mice. Degradation of the mandible was quantified with computed tomography in pre-, post-, and control cohorts, demonstrating decreased destruction of bone in the pre-cohort of animals (Fig. 3d, e).

**KBU2046 induces changes in HSP90β phosphorylation.** With the aforementioned positive phenotypic cellular and animal studies, we sought to identify the molecular basis for KBU2046's biological action. Our initial investigations were guided by our prior demonstration that low nanomolar concentrations of genistein inhibited the kinase activity of mitogen-activated protein kinase 4 (MKK4/MAP2K4/MEK4)[9], in turn inhibiting downstream phosphorylation of p38 MAPK[7] and of heat-shock protein 27 (HSP27)[14]. This translated into inhibition of MMP-2 expression and cell invasion in vitro, inhibition of human PCa metastasis in mice[8] and decreased MMP-2 expression in human prostate tissue[9]. In contrast to genistein, KBU2046 did not bind to MKK4 nor inhibit its kinase activity in vitro, and it did not inhibit downstream phosphorylation of p38 MAPK or of HSP27 in cells (Supplementary Fig. 9). This finding, while surprising, demonstrates that our chemical probe strategy de-selected for inhibition of the MKK4 signaling axis. Importantly, this provides a measure of the unbiased nature of our chemical probe strategy.

Seeking to identify KBU2046's biological target(s), we pursued alternative methods. The KinomeView® panel of antibodies (Cell Signaling Technology, Inc.) detect established protein phosphorylation motifs, and were used to probe for KBU2046-induced changes in protein phosphorylation (Fig. 4a and Supplementary Fig. 10). We prioritized phosphoprotein changes that met the following criteria: were induced in cells in vitro as well as in tumors of treated mice (from Fig. 2a), that counteracted transforming growth factor β (TGFβ)-induced effects, and that were reproducible. TGFβ is ubiquitous in vivo, is known to increase PCa cell invasion[7], and KBU2046's anti-invasion efficacy remains in spite of TGFβ-stimulated increases in cell invasion (Supplementary Fig. 11). Genistein was evaluated under identical treatment conditions for comparison. Genistein's many pharmacologic effects induced widespread changes in protein phosphorylation (Supplementary Fig. 10). In contrast, KBU2046 induced only a single change that our pre-specified criteria, i.e., a decrease in intensity of an 83 kDa protein band (blue arrow in Fig. 4a). In tumors of treated animals, KBU2046 had this same effect on this 83 kDa protein band (green arrow in Supplementary Fig. 10a). The high molecular selectivity of KBU2046 was further supported by its failure to inhibit over 400 different protein kinases and 20 phosphatases examined, in three different in vitro assay systems (Supplementary Note 2).

The 83 kDa protein was identified by pretreating PC3 cells with KBU2046 or vehicle control, treating with TGFβ and performing LC-MS/MS analysis on proteins pulled down by the Kinome-View® antibody used in Fig. 4a. Resultant data were analyzed with the SEQUEST/Sorcerer data analysis suite, and proteins further selected based upon predetermined parameters (Supplementary Fig. 12). This approach yielded a single protein, HSP90β, and indicated that KBU2046 decreased the abundance of phosphorylated Ser[226] on HSP90β by 6.6-fold (Fig. 4b and Supplementary Fig. 12).

The (S226A)-HSP90β construct lacks a Ser[226] residue, precluding phosphorylation at that site, represents a constitutive inactive mimic, and mimics the effect of KBU246 on that residue (i.e., dephosphorylation). As expected, transfection of cells with (S226A)-HSP90β inhibited cell invasion, compared to vector control (VC) transfected cells (Fig. 4c). Further, if KBU2046 were exerting efficacy by inhibiting phosphorylation of the Ser[226] residue, then removal of that residue should, by definition, preclude additional efficacy. This is exactly what is observed: in (S226A)-HSP90β transfected cells, KBU2046 does not further inhibit cell invasion, while it significantly inhibits invasion in VC cells (Fig. 4c and Supplementary Fig. 13a). The selectivity of HSP90β in mediating KBU2046 efficacy was further supported by demonstrating that small interfering RNA (siRNA)-mediated HSP90β knockdown inhibited cell invasion and abrogated KBU2046 efficacy (Supplementary Fig. 13b–d). HSP90β-specific siRNA did not knockdown HSP90α (Supplementary Fig. 13b).

| Table 1 Effect of KBU2046 and genistein on human prostate cell viability | | | | | | | | | | | | | |
|---|---|---|---|---|---|---|---|---|---|---|---|---|---|
| | | PC3M | | PC3 | | 1542NPTX | | 1542CPTX | | 1532NPTX | | 1532CPTX | |
| | | Mean[a] | se | Mean | se | Mean | se | Mean | se | Mean | se | Mean | se |
| IC20 | Genistein | 17.5 | 4.2 | 18 | 4.9 | 18.7 | 5.7 | 24.7 | 4.6 | 23 | 9.7 | 30.8 | 12 |
| | KBU2046 | NR | — | NR | — | NR | — | NR | — | NR | — | NR | — |
| IC50 | Genistein | 45.3 | 13.5 | 41.7 | 9.2 | NR | — | NR | — | NR | — | NR | — |
| | KBU2046 | NR | — | NR | — | NR | — | NR | — | NR | — | NR | — |
| Cell viability | Genistein | 46.2 | 6.3 | 39.3 | 9.3 | 80.5 | 4.4 | 77.4 | 1.2 | 66.9 | 5.3 | 71.7 | 5.8 |
| at 50 μM | KBU2046 | 100.2 | 6.1 | 88.1 | 1.4 | 107.4 | 3.6 | 103.5 | 1.3 | 95.3 | 6.2 | 99.4 | 4.8 |

NR not reached
[a]The concentrations at which KBU2046 or genistein inhibited cell growth after 3 days by 20% (IC20) and 50% (IC50) are depicted, as is the percentage of growth inhibition at 50 μM. Values are mean ± SEM from $N > 2$ separate experiments, each in replicates of $N = 3$

**Fig. 1** KBU2046 selectively inhibits cell motility. **a** Schematic flow of probe synthesis and development strategy. **b** Cell invasion. Human prostate metastatic cells (PC3, PC3-M), and HPV-transformed normal (1532NPTX, 1542NPTX) and primary cancer (1532CPTX, 1542CPTX) cells, were treated with 10 μM genistein (G), KBU2046 (46), or vehicle (CO), and after 3 days, cell invasion was measured. Values are mean ± SEM of a single experiment in replicates of $N = 4$, with similar findings in multiple separate experiments (also $N = 4$). **c** Single-cell migration. Cell migration was measured after treatment for 3 days with 10 μM KBU2046 or vehicle (control). Values are mean ± SEM of a single experiment in replicates of $N = 24$, with similar findings in a separate experiment (also $N = 24$). *Denotes Student's $t$-test $P$ value <0.05, compared to controls. **d** Human cord blood hematopoietic stem cell colony formation assay. Values are the mean ± SD number of total, CFU-GM, CFU-GEMM, or BFU-E colonies at 14 days after treatment with KBU2046, from a single experiment in replicates of $N = 2$. **e** Induction of estrogen-responsive genes. Values are the mean ± SD of a single experiment, with similar results seen in a separate experiment, both in replicates of $N = 2$

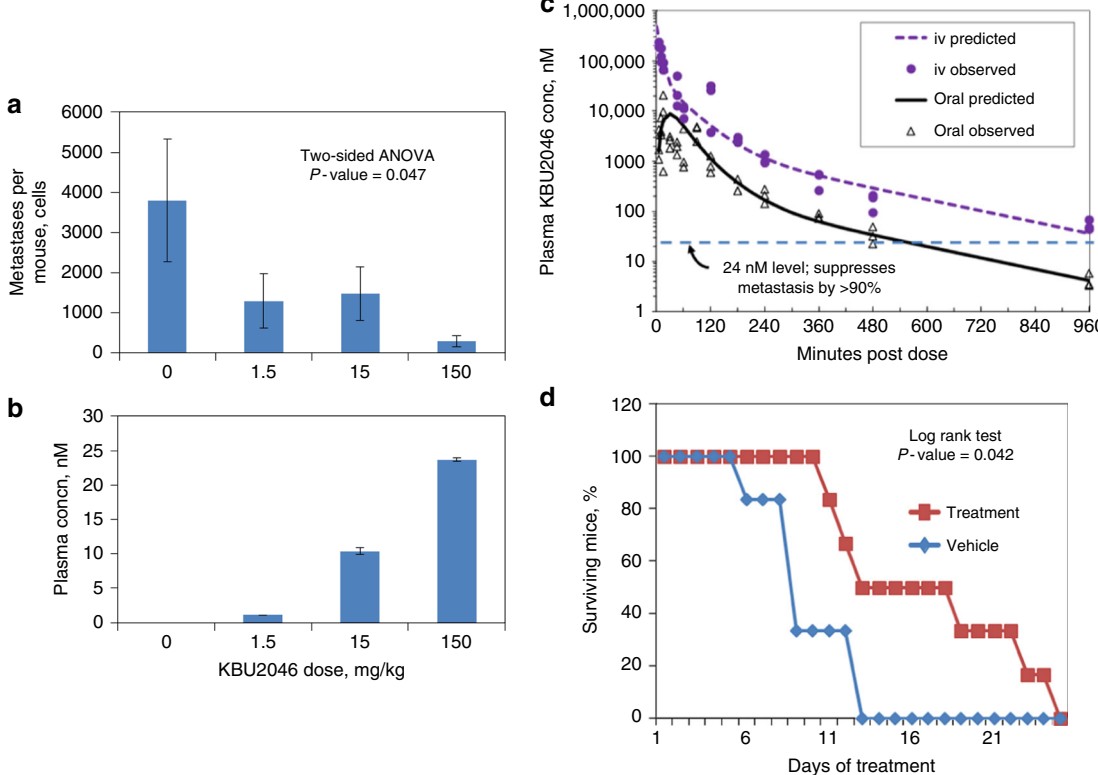

**Fig. 2** KBU2046 inhibits cancer metastasis and prolongs life. **a**, **b** Inhibition of PCa metastasis. Cohorts of $N = 12$ athymic mice bearing human PCa PC3-M cell orthotopic implants (**a**), or of $N = 5$ non-tumor bearing athymic mice (**b**), were treated with KBU2046 incorporated into chow, and resultant lung metastasis (**a**) or plasma KBU2046 concentration (**b**) measured. Values are the mean ± SEM. The relationship between dose and metastasis was evaluated by two-sided ANOVA (**a**). **c** Comprehensive characterization of KBU2046 pharmacokinetics. CD1 mice were dosed with 100 mg KBU2046/kg via oral gavage or intravenous injection (iv), and blood collected at the indicated time points (data from mice dosed at 25 mg/kg are in Supplementary Fig. 6, and corroborate 100 mg/kg findings). For each route and time point, $N = 3$ mice were sampled. Individual data points are the resultant plasma concentrations from individual mice, and are the mean of $N = 2$ measurements. The dotted horizontal line denotes a concentration of 24 nM, which was the concentration of KBU2046 measured in the blood of mice whose metastasis were suppressed by 92% (**a**, **b**). **d** Prolongation of survival in BCa-bearing mice. Mice were orthotopically implanted with human breast cancer LM2-4H2N cells, the resultant primary tumors resected, and adjuvant treatment begun with KBU2046 by daily oral gavage five times per week. The survival of $N = 6$ mice receiving vehicle was compared to that of $N = 6$ mice receiving 25 mg/kg KBU2046 by the log-rank (Mantel-Cox) test

Conversely, the pseudophosphorylated (S226D)-HSP90β construct contains a residue that provides a biological mimic of phosphorylated Ser[226], and as such constitutes a constitutively active mutant. As expected, cells transfected with (S226D)-HSP90β were more invasive than VC cells (Fig. 4d). Recognizing that pseudophosphorylated constructs only serve to mimic activated wild-type protein, it was not surprising that (S226D)-HSP90β cells were not as invasive as WT-HSP90β cells. More importantly, if KBU2046 were exerting efficacy by inhibiting phosphorylation of the Ser[226] residue, then the presence of a phospho-mimic residue should, by definition, decrease additional efficacy. This is exactly what is observed: in (S226D)-HSP90β transfected cells, KBU2046 did not significantly inhibit cell invasion, while it significantly inhibited invasion in both VC and WT-HSP90β cells (Fig. 4d and Supplementary Fig. 13). These findings demonstrate that changes in the phosphorylation status of Ser[226] on HSP90β can be altered by a small molecule, and appear to be associated with selective inhibition of cancer cell motility by KBU2046.

**KBU2046 selectively disrupts heterocomplex function.** KBU2046's effect upon HSP90β function is completely different from that of classical HSP90 inhibitors. The latter induce cytotoxicity and work by binding directly to HSP90, thereby inhibiting its enzyme activity, in turn affecting the function of large numbers of cellular kinases and other client proteins[15]. In contrast, KBU2046 was not cytotoxic and its effects on protein phosphorylation were highly specific, demonstrating a lack of effects on kinase function. HSP90β is part of a large multiprotein chaperone complex whose function involves binding a large but specific set of regulatory proteins. We reasoned that KBU2046 was changing the signature of bound client proteins, that the change was highly selective in terms of number of affected proteins, and that it was highly specific for proteins that regulate cell motility.

CDC37 is a co-chaperone that mediates the binding of over 350 client proteins to HSP90β, including over 190 kinases[16]. CDC37 is a flexible arm-like structure (protein data bank (PDB) ID: 2WOG), is highly dynamic[17], enables binding of large numbers of kinases, defines their positioning and thereby their potential to affect HSP90β phosphorylation status. We reasoned that KBU2046 was binding to either CDC37 or HSP90β, that this altered the function of the CDC37/HSP90β heterocomplex resulting in a change in the spectrum of bound client kinase proteins, that changes were highly selective and that this altered binding spectrum would in turn be responsible for KBU2046's effects upon cell motility.

There was no evidence of KBU2046 binding to either CDC37 or HSP90β by biophysical methods, inclusive of isothermal

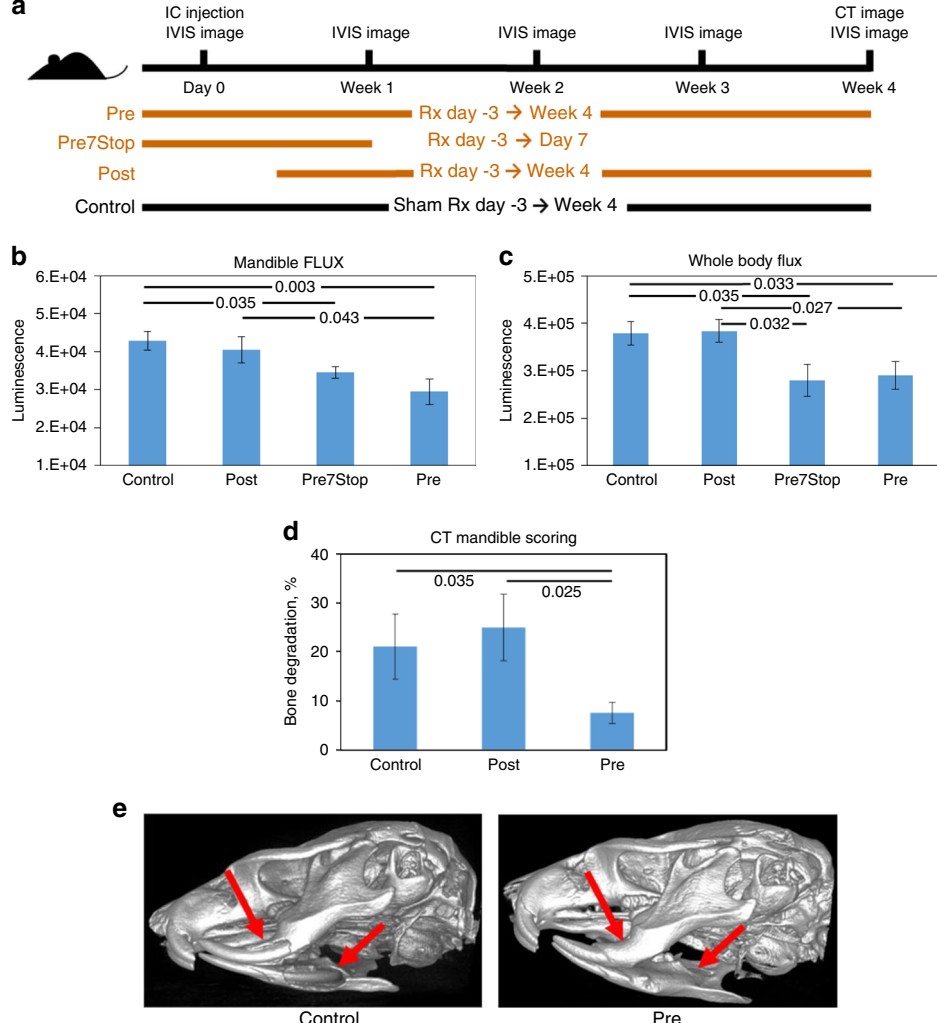

**Fig. 3** KBU2046 inhibits bone destruction. **a** Treatment schema. Athymic mice were given intracardiac (IC) injections of PC3-luc cells on day 0 under ultrasound guidance, and underwent weekly IVIS imaging starting 7 days post injection. Cohorts of $N = 20$ control, $N = 20$ pre (treatment from 3 days prior to IC injection through end of the experiment), $N = 10$ Pre7Stop (treatment from 3 days prior to IC injection through 7 days post IC injection), and $N = 10$ post-treatment (treatment from 3 days post IC injection through end of the experiment) mice were dosed with 80 mg/kg KBU2046 daily by oral gavage and mock-treated with vehicle all other times. Whole-body (**b**) and mandible (**c**) flux are depicted, as determined from weekly IVIS imaging. **d** At week 4 post injection (i.e., at the end of the experiment), CT scans were performed on control, pre-, and post-treatment cohorts, and mandibular destruction quantified. **e** Representative images of pre and control mice are depicted. Arrows denote areas of bone destruction in controls, and corresponding areas in the pre mouse. Student's $t$-test (**b**, **c**) and Fisher's exact test (**d**) $P$ values between the denoted cohorts are shown

titration calorimetry, fluorescence-based thermal shift assay, biolayer interferiometry, or by dynamic light scattering, nor by the biochemical method of drug affinity responsive target stability (DARTS) assay (Supplementary Fig. 14). DARTS provides a sensitive measure of ligand-induced changes in protein structure and dynamics by measuring the ability of a ligand to protect its target from protease digestion[18]. Although KBU2046 did not bind CDC37 or HSP90β individually, because CDC37 and HSP90β associate to form a heterocomplex[17], we went on to combine CDC37 and HSP90β in a DARTS assay, demonstrating that KBU2046 protected both proteins from digestion (Fig. 5a). The intensity of the CDC37 band increased, that of the HSP90β degradation product decreased, and both effects were statistically significant, concentration-dependent, and were evident at 10 nM. Further, the high selectivity of KBU2046 for protein binding was additionally supported by synthesizing a biotin chemical linker to KBU2046, demonstrating that it retained biological activity, that it bound to intact cells (i.e., under physiological conditions of CDC37/HSP90β heterocomplex formation), and that it failed to

bind to an array of over 9000 human proteins (Supplementary Fig. 15). Together, these findings demonstrate that KBU2046 will not bind to either CDC37 or HSP9Oβ, but that it will only bind when both proteins are present and able to form heterocomplexes. Further, all findings also indicate that KBU2046 is not acting as a classical HSP90 inhibitor. Classical HSP90 inhibitors bind isolated HSP90, without the need for co-chaperones being present, are characterized by their cytotoxic effects, are systemically toxic, particularly to the liver, and broadly inhibit client kinase protein binding, thereby exerting widespread effects upon cellular signaling and affecting a wide array of cellular processes[15]. In contrast, KBU2046 exhibits a complete lack of cellular cyctotoxicity and systemic toxicity, everts highly specific effects in both molecular-based protein phosphorylation, and cellular-based functional assays and will not bind HSP90 in isolation.

These combined experiments indicate that KBU2046 only binds to HSP90β and CDC37 when both proteins are present, does not bind to either protein alone, and together support the

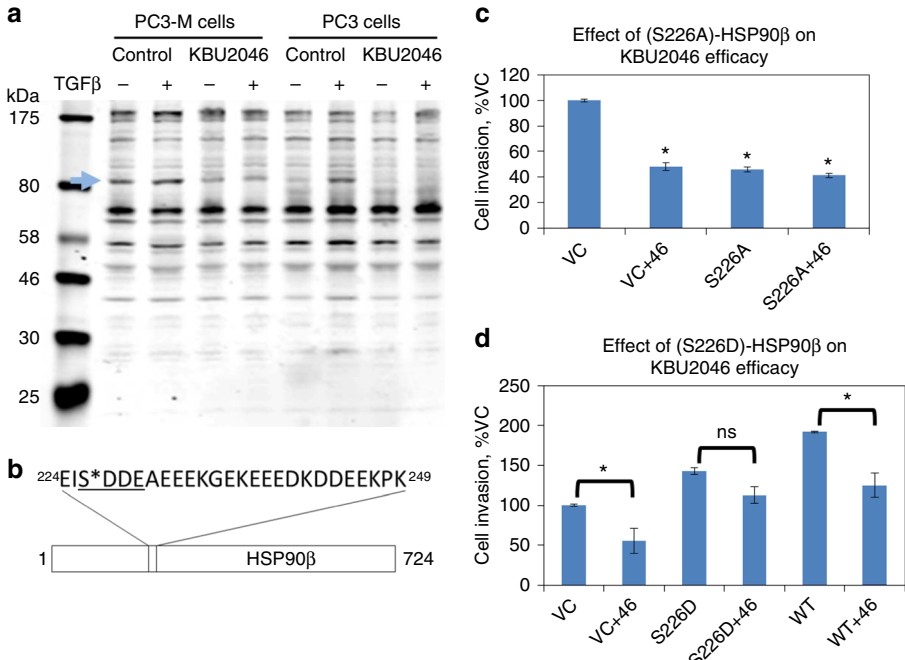

**Fig. 4** KBU2046 decreases phosphorylation of HSP90β. **a** Probing for KBU2046-induced changes in protein phosphorylation. PC3-M or PC3 cells were pre-treated with 10 μM KBU2046 for 3 days, then with ±TGFβ and the resultant cell lysate probed for changes in protein phosphorylation with the KinomeView® assay. The depicted western blot utilizes KinomeView® phospho-motif antibody, BL4176; the blue arrow denotes an 83 kDa band whose phosphorylation is inhibited by KBU2046 (see Supplementary Fig. 10 for complete KinomeView® assay screening data). **b** Proteomic analysis. PC3 cells were pre-treated with KBU2046 or vehicle, then with TGFβ, proteins from the resultant cell lysate were immunoprecipitated with BL4176, and HSP90β was identified by LC-MS/MS analysis (see Supplementary Fig. 12 for expanded proteomic assay data). The phospho-motif recognized by the antibody is underlined; S*—denotes Ser226, whose phosphorylation is decreased by KBU2046. **c**, **d** The phospho-mimetic changes in HSP90β Ser[226] structure regulate human PCa cell invasion and KBU2046 efficacy. PC3-M cells were transfected with S226A-, S226D-, or WT-HSP90β, or empty vector (VC), treated with KBU2046 or vehicle, and cell invasion measured. Values are the mean ± SEM of a representative experiment of multiple experiments (all in replicates of $N = 3$); *denotes $t$-test $P$ value <0.05 between bracketed conditions, or compared to VC

hypothesis that KBU2046 is binding in a cleft that is only present when CDC37 and HSP90β interact. A comprehensive analysis of HSP90β and CDC37 experimental structural information, including X-ray crystallographic data (PDB IDs: 1uym, 3nmq, 3pry, 2cg9, and 1us7) and chemical cross-linker physical mapping analysis[19], supports the notion that CDC37/HSP90β heterocomplex formation results in the formation of a new pocket, that is located at the interface of the two proteins. These modeling studies also predict that KBU2046 binds without any high-energy steric interactions, and with a favorable energy score (Fig. 5b–d and Supplementary Fig. 16). In this computational arrangement, Arg167 from CDC37 protrudes into a large cleft, engages in a hydrogen bond with the carboxyl side chain of Glu33 from HSP90β, which promotes the formation of a new pocket, into which KBU2046 binds.

Together, these findings suggest that KBU2046 binds the CDC37/HSP90β heterocomplex. To examine whether this is associated with an altered signature of bound client kinase proteins, we performed a modified LUMIER assay[16] to detect KBU2046-induced changes in client protein binding to CDC37/HSP90β heterocomplexes in intact cells. Of 420 kinase proteins screened, KBU2046 had highly selective effects, significantly changing the binding of only 17 (4%): binding was increased in 10 and decreased in 7 (Fig. 6a and Supplementary Fig. 17a). These findings are in contrast to classical inhibitors of HSP90 function, which have been shown, through this same assay, to affect the binding of the majority client kinase proteins[16]. Given that TGFβ increases cell motility and that KBU2046 efficacy remains in the face of TGFβ treatment (Supplementary Fig. 11), we repeated the LUMIER assay in TGFβ-treated cells, identifying 3 kinases whose

binding to complexes was significantly affected by KBU2046: RAF1 (decreased binding), RIPKI (decreased), and SGK3 (increased) (Fig. 6b and Supplementary Fig. 17a). All three proteins have been shown by others to regulate cell motility, and we demonstrate that knockdown of any one of them decreases motility (Fig. 6c, d and Supplementary Fig. 17b–d). However, only knockdown of RAF1 or of RIPK1 (i.e., the two kinases whose binding to the heterocomplex was decreased by KBU2046) mitigated KBU2046 efficacy, while KBU2046 still retained efficacy in the face of SGK3 knockdown.

DARTS assay findings (Fig. 5a) supported the notion of a direct interaction. However, the LUMIER-based approach used intact cells treated for 3 days and was unable to determine whether KBU2046 was directly interacting with heterocomplexes. Additional studies were therefore undertaken. Studies focused on RAF1. RAF1 is known to regulate cell motility and metastasis in several cancer types[20], while KBU2046's effect upon RIPK1-complex binding was minor and not further enhanced by TGFβ. KBU2046 did not alter RAF1 protein expression levels in cells (Fig. 6c). This is significant in that HSP90 inhibitors broadly inhibit chaperone activity, thereby decreasing client protein expression. We next constructed an in vitro kinase assay of purified recombinant RAF1, HSP90β, and CDC37, and demonstrated that KBU2046 decreased phosphorylation of RAF1's Ser338 activation motif (Fig. 6e). In the absence of CDC37/HSP90β heterocomplex, RAF1 activity was much lower, indicating that this effect is heterocomplex-dependent (Supplementary Fig. 18).

As KBU2046 does not directly inhibit protein kinase activity, we hypothesized that its ability to decrease HSP90β

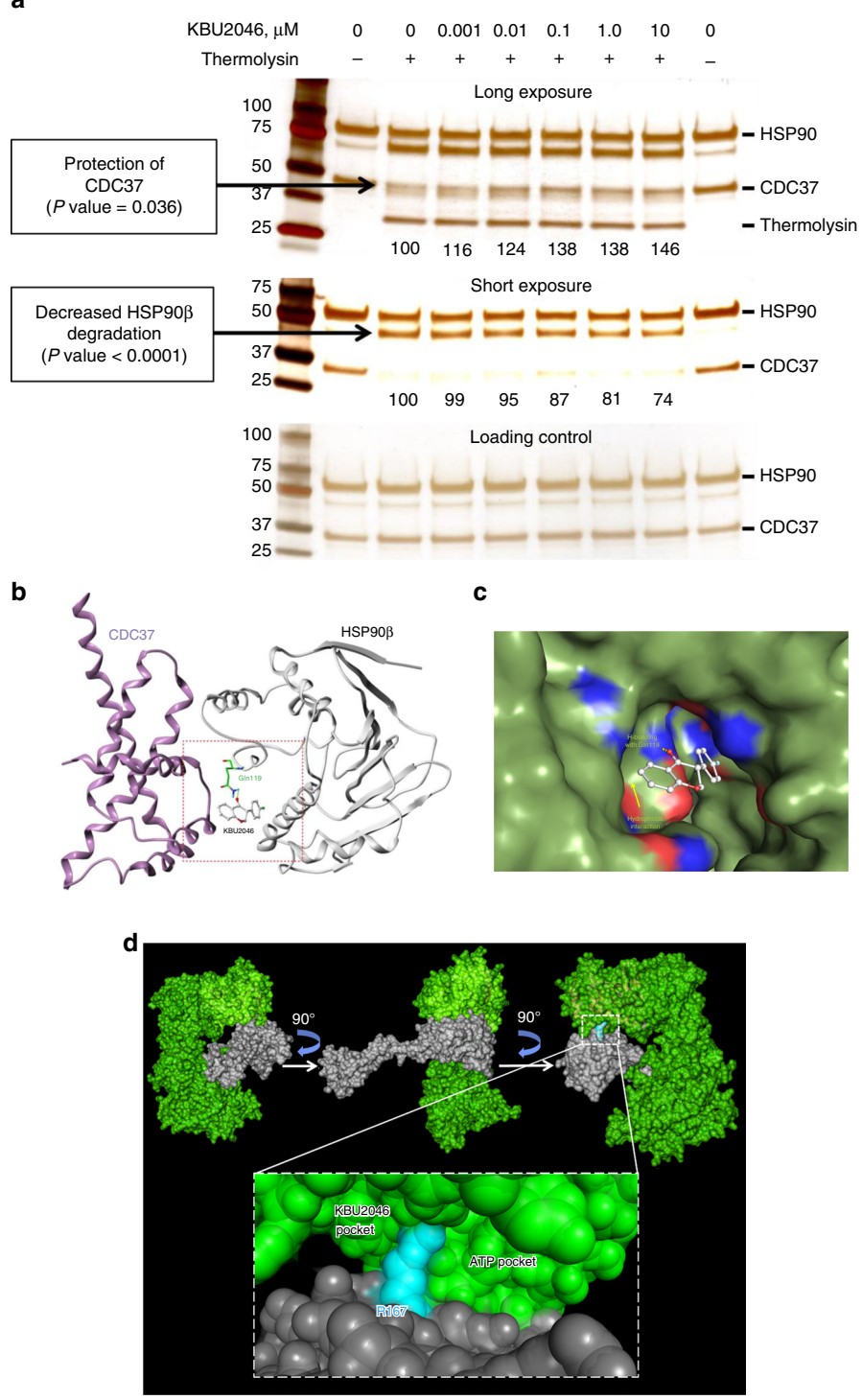

**Fig. 5** KBU2046 stabilizes CDC37/HSP90β heterocomplexes. **a** KBU2046 stabilizes HSP90β/CDC37 heterocomplexes in a DARTS assay. Equimolar amounts of HSP90β and CDC37 protein were pre-incubated with KBU2046, and resultant thermolysin reaction products were detected by silver stain following SDS-PAGE. The mean value (from $N = 3$ independent experiments) of protein bands indicated by arrows is displayed below each lane, and are expressed as the percentage of untreated control. ANOVA $P$ values for changes in band intensity with concentration are displayed. **b** In silico model of CDC37 (purple) and HSP90β (gray) depicting KBU2046 hydrogen bonding with Gln119 of HSP90β. **c** Potential surface of the computed ligand binding pocket of the CDC37/HSP90β model with KBU2046 bound. Atoms within 5 angstroms of KBU2046 are colored by element (carbon, green; nitrogen, blue; oxygen, red). **d** Potential surface of the whole CDC37/HSP90β dimer (color code: green—HSP90β; gray—CDC37; cyan—Arg167 from CDC37 bisecting the larger pocket and creating a new cleft into which KBU2046 binds)

phosphorylation resulted from changes in the signature of bound client kinases to the heterocomplex. We examined this by considering that in intact cells KBU2046 increased SGK3 binding to the heterocomplex (Fig. 6a), an effect we anticipated may in turn phosphorylate HSP90β. Utilizing our in vitro kinase assay, we demonstrated that SGK3 increased phosphorylation of

HSP90β, and that phosphorylation was further increased in the presence of KBU2046 (Fig. 6f). Recognizing the complexity of the system and the dynamic nature of the client-chaperone complex, we suspected that KBU2046-mediated inhibition of HSP90β phosphorylation was not an isolated event (i.e., not mediated by a single kinase operating in isolation). We explored this possibility

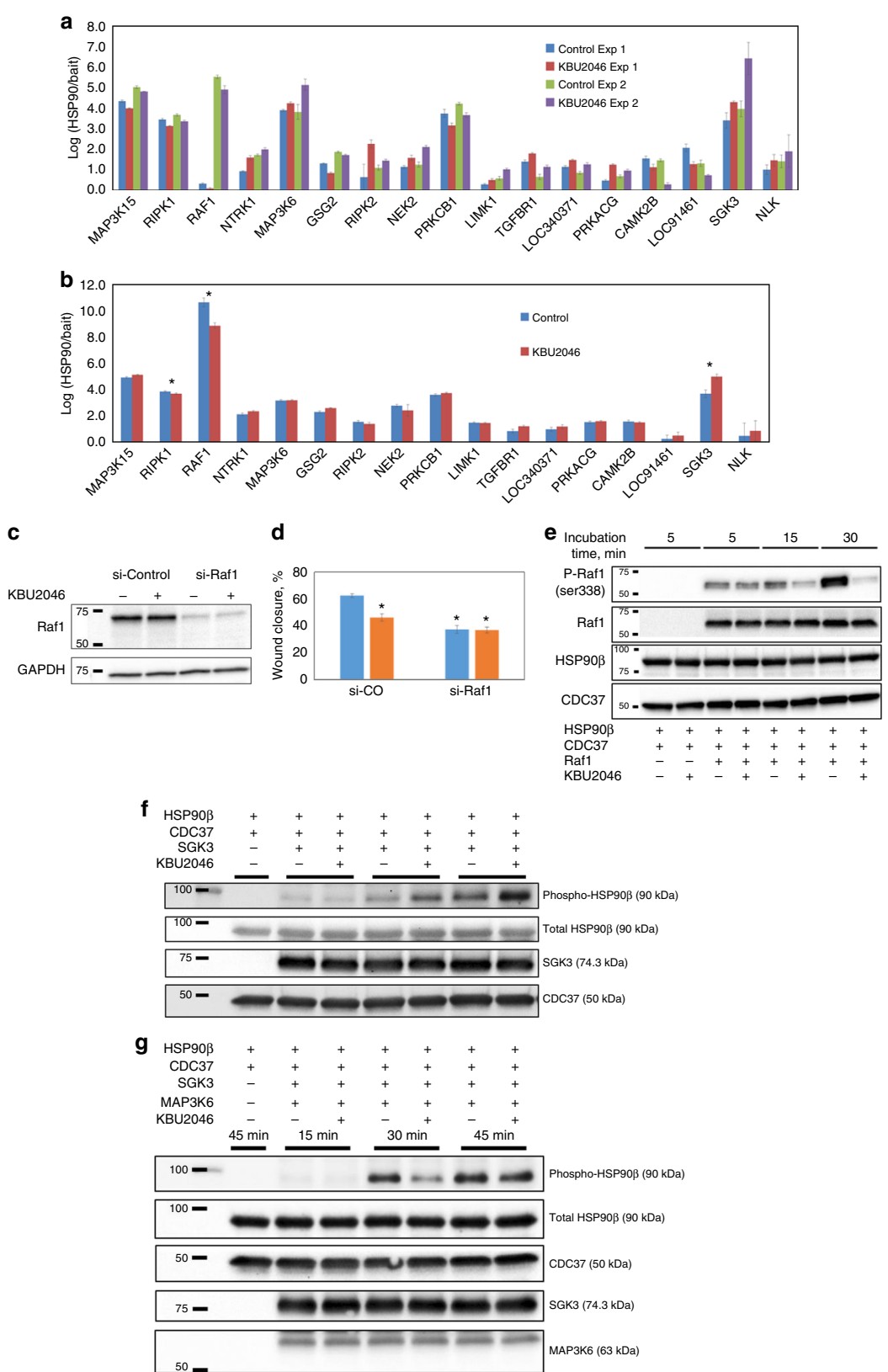

in our in vitro system by investigating the interplay of multiple kinases. In intact cells, KBU2046 increases MAP3K6 binding to complexes (Fig. 6a), but MAP3K6 is not predicted to phosphorylate the Ser[226] motif. We went on to demonstrate that when MAP3K6 is added to the in vitro kinase assay system, SGK3-mediated phosphorylation of HSP90β is not only inhibited, but in fact decreases in the presence of KBU2046 (Fig. 6g), emulating what is seen in intact cells.

## Discussion

The knowledge that increased cancer cell motility drives development of metastasis and that metastasis is responsible for the majority of cancer-related mortality has pushed the research community to identify regulators of these processes. A wide array of pathways has been shown to affect these processes. However, the identification of specific regulatory processes has been elusive, which has served as a roadblock to therapeutic modulation[4,5].

Our study deployed a natural product-inspired molecular probe strategy to address this longstanding problem. That strategy used efficient synthesis routes to generate small molecules, which were then used as biological probes. We thereby went on to demonstrate that precision modulation of cell motility could be achieved by selectively changing the signature of client kinase proteins bound to the HSP90β/CDC37 heterocomplex. Further, we demonstrate enrichment for changes in bound client kinase proteins that affect motility. We subsequently demonstrate that in the context of this altered binding signature that one of the affected client proteins, RAF1, plays an important role in mediating KBU2046 efficacy. These findings constitute a unique and specific regulatory mechanism for human cancer cell motility, and resultant downstream effects upon metastasis, end organ destruction, and survival.

In parallel with our identification of this regulatory mechanism, our studies provide for a small molecule, KBU2046, that can both serve as a biological probe and as a systemically active therapeutic. Activity is demonstrated across different cancer types and across different clinically relevant systemic models. Further, a comprehensive characterization of pharmacology and toxicity support practical therapeutic application.

Coincident with affecting the bound signature of client proteins, we also inhibit post-translational changes to HSP90β. They include a decrease in its phosphorylation status, with several of our findings pointing to phosphorylation of Ser[226] as being particularly important. We recognize that there are a relatively high number of potential phosphorylation sites on HSP90β and that the antibody used to probe phosphorylation cannot be considered specific for the Ser[226] motif. However, our investigations involving point mutations at that site and our phospoproteomics evaluation of proteins bound to the antibody do provide supportive evidence to speculate that this site is of regulatory importance.

Of high importance, KBU2046 lacked all of the hallmarks of classical HSP90 inhibitors. Specifically, KBU2046 was not cytotoxic, lacked systemic toxicity, did not decrease expression of client proteins, and it did not broadly alter kinase function. Further, KBU2046 only bound to HSP90β and CDC37 when both proteins were present and able to form heterocomplexes, and would not bind to HSP90β as an isolated protein nor to CDC37 as an isolated protein. This is in contrast to classical HSP90 inhibitors, which are characterized by their ability to bind HSP90 as an isolated protein.

We go on to present a structural model in which KBU2046 binds within a cleft that is created through the binding of HSP90 and CDC37, and exists at the interface between these two proteins. While our model integrates information from physical mapping, biochemical analysis, and crystallographic structure, its final construct was in silico, and it should therefore be considered speculative. Several other models also describe the structure of HSP90/CDC37 interactions[21,22]. Those models differ from each other, and from ours. For example, our model integrated information from physical mapping (based on chemical cross linkers), from structural information reported in the literature, from our findings implicating KBU2046 interaction with HSP90/CDC37 heterocomplexes, and only took into consideration heterocomplex structure (i.e., in the absence of bound client proteins). In contrast, one recent model used cyroEM to evaluate HSP90/CDC37 heterocomplex structure in the context of its binding to the CDK4 client protein[21]. There are many other potential explanations. It is likely that each model describes a particular state, and different states are possible. Our model should therefore be considered a speculative framework from which future studies can be launched. Future investigations will need to be performed in order to determine the effects of bound vs. unbound client protein, different client proteins, binding of small molecules to the HSP90/CDC37 heterocomplex structure, and the ultimate validity of each of these models.

From our findings, we propose an integrated structural and functional model. KBU2046 binds to a cleft that is formed when HSP90β and CDC37 bind to form a heterocomplex. This in turn affects the ability of the heterocomplex to bind client kinase proteins in a very precise manner, selectively affecting those that regulate cell motility. Of these, RAF1 is of particular importance. KBU2046 decreases RAF1 binding to the heterocomplex, resulting in decreased activation, and inhibition of cell motility. KBU2046's precision type of effect on chaperone function differentiates it from classical HSP90 inhibitors, which broadly disrupt client protein function, and underlie KBU2046's lack of toxicity and selective modulation of cell motility.

Together, the findings of this study provide a rational platform to move investigations into humans. That platform includes mechanistic strategy and the physical tools with which to affect it. Also, this study provides proof of principle findings that through pharmacologic means it is possible to induce precision

**Fig. 6** KBU2046-mediated changes in the signature of client proteins bound to the HSP90β/CDC37 heterocomplex mediate effects upon cell motility. **a** LUMIER assay. HEK293T cells were transfected with 1 of 420 different protein kinases, treated with 10 µM KBU2046 ($N = 5$ replicates) or vehicle control ($N = 5$) for 3 days, and LUMIER assays performed. The $N = 17$ kinases that gave significant findings (Student's $t$-test <0.05) in the same direction in each of two separate experiments are depicted. Each separate treatment and kinase condition in each of two separate experiments was conducted in replicates of $N = 5$. **b** The experiment was then repeated for these 17 kinases in the presence of TGFβ treatment, and those demonstrating significant differences ($t$-test <0.05) in the same direction as in **a** are denoted by *. **c**, **d** Wound healing assay. PC3 cells were transfected with siRNA targeting RAF1 (si-Raf1) or non-targeting siRNA (si-control), treated with KBU2046 or vehicle as above, and RAF1 protein measured by western blot (**c**) and effects upon wound healing measured (**d**). **e** Inhibition of RAF1. Purified recombinant HSP90β, CDC37, and RAF1 were combined with KBU2046, as indicated, incubated in an in vitro kinase assay for the indicated times, and western blot for RAF1-Ser[338] phosphorylation performed. **f**, **g** Effect on HSP90β/CDC37 heterocomplex formation and function in vitro. Purified recombinant HSP90β, CDC37, RAF1, SGK3, or MAP3K6 were combined and treated with KBU2046 or vehicle control, as indicated, incubated in an in vitro kinase assay for the indicated times, and western blot performed, as denoted. All experiments were repeated at separate times at least once, with similar results

modulation of the signature of client protein binding to chaperone scaffold proteins, in turn resulting in highly selective functional effects at the cellular and systemic level. In parallel, this approach serves to inform us about novel pathways for regulating critical biological processes. Finally, our approach that coupled efficient chemical synthesis routes with a well-designed phenotypic screening strategy has the potential to be broadly applied as a tool to interrogate other critical biological processes.

## Methods

**Chemical synthesis.** Procedure for Large-Scale Production of 4′-fluoroisoflavanone (KBU2046):

3-(dimethylamino)-1-(2-hydroxyphenyl)prop-2-en-1-one.

The starting materials 2′-hydroxyacetophenone (50 mmol, 6.02 mL) and $N,N$-dimethylformamide dimethyl acetal (50 mmol, 6.64 mL) were added to a 10–20 mL microwave vial. The vial was capped and heated in a Biotage Initiator microwave synthesizer at 150 °C and 11 bar for 10 min. The resulting dark orange liquid was allowed to cool to 23 °C, at which time yellow–orange crystals crashed out of solution. The crystals were collected and washed with hexanes (50 mL), then dried and weighed to give 3-(dimethylamino)-1-(2-hydroxyphenyl)prop-2-en-1-one (9.09 g, 95%) as orange–yellow needles. Analytical data for 3-(dimethylamino)-1-(2-hydroxyphenyl) prop-2-en-1-one: $^1$H nuclear magnetic resonance (NMR) (500 MHz, CDCl$_3$) δ 13.97 (s, 1 H), 7.92 (d, $J = 12.1$ Hz, 1 H), 7.72 (dd, $J = 8.0$, 1.6 Hz, 1 H), 7.38 (ddd, $J = 8.5$, 7.2, 1.6 Hz, 1 H), 6.96 (dd, $J = 8.3$, 1.2 Hz, 1 H), 6.85 (ddd, $J = 8.3$, 7.2, 1.2 Hz, 1 H), 5.81 (d, $J = 12.1$ Hz, 1 H), 3.23 (s, 3 H), 3.01 (s, 3 H); $^{13}$C NMR (126 MHz, CDCl$_3$): δ 191.5, 163.0, 154.8, 134.0, 128.2, 120.3, 118.3, 118.0, 90.1, 45.5, 37.5; ultra performance liquid chromatography/mass spectrometry (UPLCMS): mass calculated for C$_{11}$H$_{13}$NO$_2$, [M + H]$^+$, 192. Found 192.

3-bromochromone.

3-bromochromone was prepared by a procedure taken from Gammill[23]. To a flame-dried 250 mL round bottom flask, was added 3-(dimethylamino)-1-(2-hydroxyphenyl)prop-2-en-1-one (36.6 mmol, 7.0 g), which was dissolved in CHCl$_3$ (70 mL). The reaction flask was cooled to 0 °C in an ice bath, then Br$_2$ (36.6 mmol, 1.87 mL) was added dropwise through an addition funnel. After all of the Br$_2$ was added, water (70 mL) was added slowly to the reaction and it was stirred at 23 °C for 10 min. The dark orange–yellow organic layer was then separated from the aqueous layer, which was back-extracted with 3 × 50 mL CHCl$_3$. The combined organic layers were dried over Na$_2$SO$_4$ and concentrated to give a dark orange oil. This was purified by flash column chromatography (SiO$_2$, 10% EtOAc/hexanes) to afford 3-bromochromone (5.26 g, 64%) as an off-white solid. Analytical data for 3-bromochromone: $^1$H NMR (500 MHz, CDCl$_3$) δ 8.31 (dd, $J = 8.0$, 1.7 Hz, 1 H), 8.27 (s, 1 H), 7.75 (ddd, $J = 8.7$, 7.1, 1.7 Hz, 1 H), 7.57–7.44 (m, 2 H); $^{13}$C NMR (126 MHz, CDCl$_3$): δ 172.3, 156.1, 153.8, 134.2, 126.5, 125.9, 123.2, 118.2, 110.7; UPLCMS: mass calculated for C$_9$H$_5$BrO$_2$, [M + H]$^+$, 226. Found 226.

Palladium tetrakis (triphenylphosphine) (Pd(PPh$_3$)$_4$). The catalyst for the Suzuki-Miyaura cross-coupling reaction to synthesize 4′-fluoroisoflavone was made using a procedure by Coulson[24]. To a flame-dried 100 mL Schlenk flask was added PdCl$_2$ (5 mmol, 890 mg) and triphenylphosphine (25 mmol, 6.56 g). The solids were dissolved in DMSO (60 mL), then the mixture was purged with N$_2$ and heated to 145 °C, at which time it turned a bright yellow–orange color. The reaction was removed from heat and allowed to stir at room temperature for 15 min, then hydrazine hydrate (20 mmol, 0.972 mL) was added via syringe, with a vent needle in place to account for the formation of N$_2$ gas. After the hydrazine hydrate had been added, the reaction was cooled to 23 °C, during which time a yellow solid crashed out of solution. The solid was washed under Schlenk filtration conditions with 2 × 50 mL EtOH, then 2 × 50 mL ether to yield Pd(PPh$_3$)$_4$ (5.31 g, 94%) as a canary yellow solid that was stored under N$_2$ in the glovebox.

4′-fluoroisoflavone.

4′-fluoroisoflavone was prepared on large scale according to a procedure from Suzuki and Miyaura[25]. To a flame-dried 500 mL round bottom flask was added 3-bromochromone (50 mmol, 11.25 g), 4-fluorophenylboronic acid (55 mmol, 7.69 g), and Na$_2$CO$_3$ (100 mmol, 10.6 g). The solids were dissolved in a mixture of benzene (100 mL) and water (50 mL), and the system was purged with N$_2$ for 10–15 min. The Pd(PPh$_3$)$_4$ catalyst (2.5 mmol, 2.89 g) was then added, at which time the reaction turned a bright orange. The flask was equipped with a reflux condenser and the reaction was heated to reflux (80 °C) overnight. After ~16 h, the reaction was cooled to 23 °C and was diluted with EtOAc (250 mL), then the crude material

was passed through a plug of silica with EtOAc as the eluent. The organic material was dried over Na$_2$SO$_4$ and concentrated to give a dark brown solid that was adsorbed onto silica gel using DCM. Material purified by flash column chromatography (SiO$_2$, 20% EtOAc/hexanes) to afford 4′-fluoroisoflavone (8.14 g, 67% yield) as a yellow–orange solid that showed minor impurities by $^1$H NMR spectroscopy. Slightly impure material was taken onto the next step of the synthesis without further purification. $^1$H NMR (500 MHz, CDCl$_3$) δ 8.35 (dd, $J = 8.0$, 1.6 Hz, 1 H), 8.05 (s, 1 H), 7.73 (ddd, $J = 8.7$, 7.1, 1.7 Hz, 1 H), 7.61–7.54 (m, 2 H), 7.53 (dd, $J = 8.4$, 1.1 Hz, 1 H), 7.48 (ddd, $J = 8.2$, 7.0, 1.1 Hz, 1 H), 7.17 (ap t, $J = 8.7$ Hz, 2 H); $^{13}$C NMR (126 MHz, CDCl$_3$): δ 176.2, 163.8, 161.8, 156.2, 152.9, 133.8, 130.7, 127.8, 126.4, 125.4, 124.5, 118.1, 115.5; UPLCMS: mass calculated for C$_{15}$H$_9$FO$_2$, [M + H]$^+$, 241. Found 241.

4′-fluoroisoflavanone (KBU2046).

The reaction conditions to synthesize 4′-fluoroisoflavanone on large scale were adapted from a procedure reported by Wähälä[26]. To a flame-dried 500 mL round bottom flask was added 4′-fluoroisoflavone (25 mmol, 6.01 g), and the solid was dissolved in dry THF (100 mL). The solution was cooled to −78 °C (dry ice/acetone bath), monitored by a thermocouple. Once the solution had cooled to the desired temperature, L-selectride (55 mmol, 55 mL, 1 M solution in THF) was added dropwise over a period of 30–45 min. The reaction was then allowed to stir at −78 °C for 2 h, after which time it was quenched with MeOH (55 mL) at −78 °C. The mixture was then poured into 300 mL of water, and the aqueous layer was adjusted to pH 7 with 2 M HCl. The aqueous layer was extracted 2 × 200 mL with EtOAc, then the combined organic layers were dried over Na$_2$SO$_4$ and concentrated to give a dark brown oily solid. This was purified by flash column chromatography (SiO$_2$, 1:1 hexanes:DCM) to give 4.5 g of crude material that was recrystallized in hexanes to afford 4′-fluoroisoflavanone (3.4 g, 56%) as a fluffy white solid. It was checked for purity by both $^1$H NMR and HPLC analysis, with material that was >98% pure taken onto animal studies. Analytical data for isoflavanone 4′-fluoroisoflavanone: $^1$H NMR (500 MHz, CDCl$_3$) δ 7.98 (dd, $J = 7.9$, 1.7 Hz, 1 H), 7.55 (ddd, $J = 8.6$, 7.1, 1.7 Hz, 1 H), 7.33–7.24 (m, 1 H), 7.14–6.99 (m, 4 H), 4.77–4.54 (m, 2 H), 4.02 (dd, $J = 9.0$, 5.3 Hz, 1 H); $^{13}$C NMR (126 MHz, CDCl$_3$) δ 192.0, 163.3, 161.5, 136.2, 130.7, 130.2, 127.8, 121.8, 120.9, 117.9, 115.8, 71.4, 51.5; UPLCMS: mass calculated for C$_{15}$H$_{11}$FO$_2$, [M + H]$^+$, 243. Found 243.

Synthesis of additional compounds. A series of related analog compounds was synthesized in addition to the parent 4′-fluoroisoflavanone (KBU2046). These compounds were prepared in the same general manner of KBU2046 and the structures of each such compound are depicted in supplementary figures. The structure and purity of the additional analogs were confirmed by NMR spectroscopy ($^1$H and 13 C) as well as by UPLCMS (minimal ion fragmentation). All compounds were isolated and stored in powdered form (in the absence of light) and were formulated into DSMO stock solutions just prior to use.

**Cell culture and reagents.** PCa (PC3, LNCaP, and DU145), breast cancer (MDA-MB-231 and MCF-7), colon cancer (HCT110 and HT29), and lung cancer cells (H226 and A549) were obtained from American Type Culture Collection and not further authenticated. The origin, characteristics, for PC3-M, as well as for human papilloma virus (HPV) transformed primary 1532NPTX (normal), 1532CPTX (cancer), 1542NPTX (normal), and 1542CP3TX (cancer) cell lines, have previously been described by us[27]. The origin of the stable polyclonal HEK293T cell lines expressing Renilla-HSP90β were previously described[16]. LM2-4H2N human breast cancer metastatic variant cells were derived from MDA-MB-231 cells as described[28], and the tdTomato-Luc2-expressing cell line was established by transduction of these cells with a lentiviral vector encoding fluorescent (tdTomato) and bioluminescent (Luc2) genes as described[29]. All cells were cultured as described[27,28], were maintained at 37 °C in a humidified atmosphere of 5% carbon dioxide with biweekly media changes, were drawn from stored stock cells, and replenished on a standardized periodic basis and were routinely monitored for *Mycoplasma* (PlasmoTest™, InvivoGen, San Diego, CA), at least every 3 months. Cells were authenticated by the following: they were acquired from the originator of that line, grown under quarantine conditions, expanded and stored as primary stocks and not used until following conditions were met: mycoplasma negative; through morphologic examination; growth characteristics; hormone responsiveness or lack thereof, when applicable; replenished from primary stocks at least every 3 months; working with a single primary stock cell line at a time with hood sterilization in between.

Phospho-HSP27 (catalog #2401), phospho-p38 MAPK (#4631), p38 MAPK (#9212), phospho-CK2 substrate (#8738), CDC37 (#3618), HSP90β (#5087), GST (#2622), phsopho-c-RAF (ser338) (#9427), SGK3 (#8573), GAPDH (#2118), anti-mouse IgG-HRP linked secondary (#7076), and anti-rabbit IgG-HRP linked secondary (#7074) antibodies were purchased from Cell Signaling Technology. MAP3K6 (#SAB1300114) antibody, estradiol, and 4′,5,7-trihydroxyisoflavone, genistein, were purchased from Sigma-Aldrich. Pierce ECL western blotting substrate (#32106) and SuperSignal West Femto maximum sensitivity substrate (#34096) were purchased from Thermo Scientific. All primary antibodies were used at a dilution of 1:1000 and corresponding secondary antibodies used at a dilution of

1:5000. The following recombinant proteins were purchased: RAF1 (EMD Millipore; #17-360), MAP3K6 (Abnova; #P5592), and SGK3 (Thermo Scientific; #PV3859).

**Cell invasion assays**. Boyden chamber cell invasion assays were performed as previously described by us[30], using either denatured collagen (BD Biosciences) or type IV human collagen (BD Biosciences), with all experiments repeated, each in $N = 4$ replicates. In some experiments, as indicated, cells were transfected with an expression plasmid, using Lipofectamine 2000™ (Invitrogen), or with siRNA, using DharmaFect Duo (Thermo Scientific) and co-transfected with β-Galactosidase (Plasmid pCMV•SPORT-βgal; Life Technology).

**Cell growth inhibition assays**. Three-day MTT cell growth inhibition assays were performed as described by us[31]. Assays were in replicates of $N = 3$, and were repeated.

**Western blots**. Western blots were performed as described by us[32]. All western blots were repeated at least once. Uncropped scans of the most important blots are depicted in Supplementary Fig. 19.

**Cell migration assays**. Single-cell motility assays were conducted by adding $10^4$ cells to 35 mm tissue culture dishes (BD Falcon) coated with collagen I (BD Biosciences), incubating at 37 °C in 5% $CO_2$, performing time-lapse imaging using a Biostation (Nikon Instruments), tracking the path of $N \geq 35$ cells using ImageJ software and the Manual Tracker plug-in, and using Chemotaxis and Migration Tool plug-in for data analysis.

**Scratch wound assays**. Cells were transfected with the indicated siRNA constructs per manufacture protocol (GE-Dharmacon), cultured 48 h with 10 μM KBU2046 or vehicle, and scratch wound assay performed as described by us[33]. All experiments were conducted in $N = 4$ replicates, and repeated.

**Constructs, transfection, and luciferase assays**. Constructs were purchased or gifts: constitutive active MEK4EE (MAP2K4-EE; residues 37-399; Addgene, plasmid 14813), pRL-TK-Renilla luciferase (Promega), pCMV-β-galactosidase (Agilent Technologies), and pcDNA-GFP (Invitrogen), HSP90β was from Pawel Bieganowski (Mossakowski Medical Research Center PAS, Poland)[34], estrogen-responsive promoter-luciferase reporter construct, pERE-Luc, was from Craig Jordan (Georgetown University)[35], human CDC37 in pET15b plasmid was from Avrom Caplan (City College of New York)[36]. siRNA used Dharmacon ON-Targetplus SMARTpool™ siRNA directed against HSP90β (cat # L-005187-00-0010) non-targeting siRNA (cat # D-001810-10-05) used TransIT-LT1 Transfection Reagent (Mirus Bio LLC), or with Dharmafect Duo (Thermo Scientific, Lafayette, CO) for co-delivery of plasmid. Luciferase assays were performed as described by us[30].

**Animal models of metastasis and of systemic effects**. All animal studies adhered to the NIH Guide for the Care and Use of Laboratory Animals, were treated under institutional IACUC-approved protocols by Oregon Health and Sciences University and Northwestern University, complied with all federal, state, and local ethical regulations, housed in barrier (for immunocompromised mice) or conventional facilities, with a 12-h light/dark cycle and given soy-free food and water ad libitum. Animal study sample size determination: sample sizes were determined using the samples size estimation formula for differences in means with power set 80%, two-sided $\alpha = 0.05$, and a pre-specified effect size of 30%.

Prostate cancer: orthotopic implantation: Orthotopic implantation of human GFP-PC3-M PCa cells into 6–8 week male Balb/c athymic mice (Charles River Laboratories) and quantification of distant metastasis was performed as described by us[37]. Treatment with KBU2046, incorporated into Harlan Teklad 2016S® chow, began 1 week prior to implantation. Animals were excluded from the analysis if they died and/or met the criteria for killing in the 7 day post-operative period. Experimental groups were randomly assigned to cages prior to the initiation of the study. Metastasis were scored in a blinded fashion. Specifically, animals were assigned an ID number, resultant histologic slides contained a separate pathological ID number, slides were scored in a random fashion, after which the two numbers were linked up.

Breast cancer: orthotopic implantation: Orthotopic implantation of $2 \times 10^6$ dTomato-LM2-4H2N cells in matrigel:PBS human breast cancer cells into 5–6 week old female SCID-Beige mice (Taconic), followed by resection of resultant primary tumor, provides a model wherein survival is dictated by metastatic burden, and was performed as described[12]. KBU2046 treatment by oral gavage 5 days per week began 4 weeks after resection, with weekly IVIS imaging. Animals were excluded from the analysis if they died and/or met the criteria for euthanasia in the 7 day post-operative period. Experimental groups were randomly assigned based upon size of primary tumor before treatment, in and manner that ensured equal distribution of tumor sizes across treatment groups.

Prostate cancer: intracardiac (IC) injection: IC injection of $4.0 \times 10^5$ PC3-Luc cells into male 6–8 week old athymic mice with IVIS imaging was performed as described[13], and was done so under ultrasound guidance (Supplementary Fig. 9). Mice were treated pre- and/or post-IC injection, as indicated. The pretreatment cohort emulates a metastasis naive model wherein cell motility is required in order for cells to invade into a distant organ, in this case, the jaw bone, for which this model is widely used. If KBU2046 were inhibiting cell motility, then in this model it would act to inhibit metastasis formation. In contrast, with the post-implantation cohort, cells have already completed invasion into the jaw bone, and distant metastasis have been established. If KBU2046 were only inhibiting cell motility, then in this model, it should exhibit no efficacy. IVIS imaging was performed 30 min after IC injection to confirm systemic distribution of cells, and thereafter weekly for 4 weeks starting 7 days post injection, allowing real-time tracking of metastasis development and growth. Computed tomograpgy (CT) (Inveon, Seimens) radiographic imaging was performed on some mice, as indicated, and resultant images analyzed in a blinded fashion using Inveon Research Workplace 4.2 visualization/analysis software and ImageJ. Animals were excluded from the analysis if 30 min post-injection IVIS imaging revealed a focal signal in the chest indicating a failed injection where cells were not distributed into circulation. Experimental groups were randomly assigned to cages prior to the initiation of the study. CT images were analyzed in a blinded fashion to determine total bone loss using both the Invenon Research Workplace 4.2 visualization/analysis software and ImageJ. Specifically, to asses bone loss in an unbiased manner, an operator randomly loaded images into the system and blocked out identifying information prior to analysis by two separate blinded individuals.

Systemic effects: Off-target effects were sought by performing histologic examination of tissue and by measuring organ function.

Histologic examination of tissue: Organs of athymic male mice stained with hematoxylin and eosin, trichrome, or giemsa were microscopically examined by a mouse pathologist (L.L.) in a blinded fashion, and toxicity scored using an established semi-quantitative histological scoring system[38] (Supplementary Fig. 8). The following clinical chemistry parameters were measured in blood of CD1 male and female mice by Charles River Research Animal Diagnostic Services: cholesterol, triglycerides, alanine aminotransferase, aspartate aminotransferase, total bilirubin, glucose, phosphorus, total protein, calcium, blood urea nitrogen (BUN), creatinine, albumin, Na, K, Cl, white blood cells (with differential), red blood cells, hemoglobin, and platelets.

**Hematopoietic stem cell colony formation assay**. Fourteen-day colony formation assays were conducted as described by us[39], using human cord blood CD34+ stem cells (AllCells Inc.), using MethoCultExpress™ colony growth media (StemCell Technologies Inc.), performed in replicates of $N = 2$.

**KBU2046 quantification and pharmacokinetics**. For determination of pharmacokinetic (PK) parameters, KBU2046 was administered by either intravenous injection or oral gavage to groups of $N = 3$ CD1(ICR) mice at 0 (i.e., vehicle only), 25, or 100 mg/kg. Blood was collected into EDTA by terminal cardiac puncture before drug administration (i.e., baseline) and at 5, 10, 15, 30, 45, 60, 90, 120, 180, 240, 360, 480, 960, 1440 min after administration. In separate experiments, as described above, Balb/c athymic mice were dosed with KBU2046 incorporated into chow for 35 days, after which blood was collected by terminal cardiac puncture. Resultant plasma (~250 μl/mouse) was stored at −80 °C.

Plasma KBU2046 concentrations were measured in duplicate by liquid chromatography-tandem mass spectrometry after sample preparation by solid-phase extraction. Specifically, 0.1 mL of a sample, 10 μL of 0.1 μg/mL internal standard solution (3-(2-chlorophenyl)-(4H-1-benzopyran-4-one, a chloride analog of KBU2046), 3800 μL of water, and 10 μl of 85% phosphoric acid were added, vortexed, and stored at 4 °C for 2 h. After washing a 96-well Strata-X 33 μm polymeric reversed phase 30 mg/well solid-phase extraction plate (Phenomenex) with methanol and water, sample was applied, washed with 20% methanol in water, eluted with 70% acetonitrile/30% methanol, dried at 50 °C under $N_2$, reconstituted with 100 μL of mobile phase, and 20 μL was analyzed on an API 3000 liquid chromatography-tandem mass spectrometry system (Applied Biosystems) with an Agilent 1100 series HPLC system (Agilent Technologies). Samples were eluted isocratically from a Synergi 4-μm MAX-RP 100 Å column (2.0 × 50 mm; Phenomenex) by a mobile phase consisting of 10 mM ammonium formate in water and methanol (30:70 [vol/vol]) at a flow rate of 0.20 mL/min. The tandem mass spectrometer was operated with its electrospray source in the positive ionization mode. The mass-to-charge ratios of the precursor-to-product ion reactions monitored were 243.5 → 125.1 for KBU2046 and 257.0 → 165.0 for the internal standard. The retention time of KBU2046 was ~2.7 min while that of the internal standard was ~2.3 min. The linear range for plasma KBU2046 standard curves was 0.1–25.0 ng/mL, with coefficients of variation of 10% or less throughout the entire concentration range. Fresh plasma standard curves were prepared in blank plasma and run on the day of analysis of plasma samples.

Plasma KBU2046 concentration vs. time relationships after both intravenous and oral drug administration were modeled simultaneously using the SAAM II software system (SAAM Institute), implemented on a WindowsTM-based PC (see PK modeling schema). Plasma concentrations were modeled with a three-compartment PK model using a naive pooled data approach[40]. Oral drug absorption was characterized by a tanks-in-series delay element, to account for the non-instantaneous appearance of drug in the body. Simultaneous estimation of PK

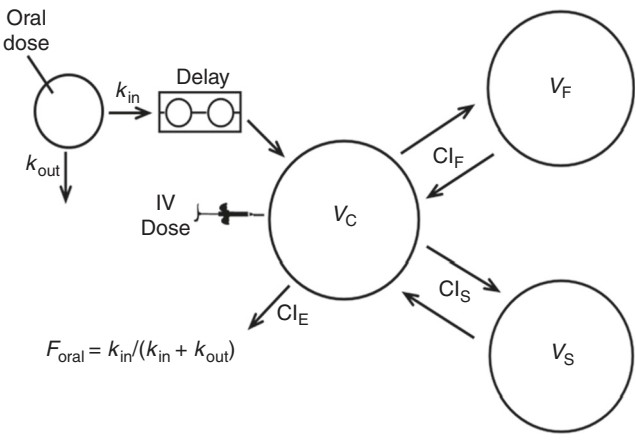

PK modeling schema

**Fig. 7** Pharmacokinetic model of the disposition of orally and intravenously administered KBU2046. Schematic depiction of the three-compartment pharmacokinetic (PK) model used determine plasma KBU2046 concentration versus time relationships after intravenous (IV) or oral administration. The rate constants $k_{in}$ and $k_{out}$ determine the fraction of the drug absorbed after oral administration (bioavailability, $F_{oral}$). Drug enters the central compartment (central volume, $V_C$) from which it equilibrates with the rapidly equilibrating (fast volume, $V_F$) and slowly equilibrating (slow volume, $V_S$) peripheral compartments at rates dictated by their respective intercompartmental clearances ($Ci_F$ and $Cl_S$). Drug is eliminated from VC by elimination clearance ($Cl_E$).

model parameters for both routes of administration permitted estimation of the bioavailability of the orally administered drug (F)[41]. The SAAM II objective function used was the extended least-squares maximum likelihood function using data weighted with the inverse of the model-based variance of the data at the observation times[42]. Model misspecification was sought by visual inspection of the measured and predicted marker concentrations vs. time relationships (Fig. 7)[42].

**Quantitative reverse transcriptase PCR**. RNA was isolated and qRT-PCR was performed as described by us[43], analyzed by the $2^{-\Delta\Delta Ct}$ method[44], using primer/probes sets (ABI), HSP90α (Hs00743767_sH), HSP90β (Hs00427665_g1), trefoil factor 1 (TFF1; Hs00907239_m1), cathepsin D (CTSD; Hs00157205_m1), progesterone receptor (PGR; Hs01556702_m1), and GAPDH (Hs99999905_m1). Assays were repeated, each in replicates of N = 2.

**Biophysical and biochemical binding assays**. Fluorescent thermal shift[45], isothermal titration calorimetry[19], and biolayer interferometry assays[46] were all performed as described by us using MEK4EE (37-399, S257E, T261E) cloned into pMCSG7 and KRX Competent E. coli cells (Promega Inc.) and above denoted full-length HSP90B and CDC37 vectors and Rosetta BL-21 and BL-21 DE-3 competent E. coli cells, respectively. DARTS assays were conducted as described[18], using equimolar amounts of CDC37 and HSP90β, thermolysin digestion, and silver stain visualization (ProteoSilver Silver Stain Kit, Sigma-Aldrich).

**In vitro kinase assays**. MEK4/MKK4/MAP2K4 in vitro kinase assay was performed as described by us[45]. HSP90β/CDC37 heterocomplex kinase assays with RAF1, SGK3, and MAP3K6 assays incubated indicated proteins in 20 mM MOPS, pH 7.2, 25 mM β-glycerol phosphate, 5 mM EGTA, 1 mM sodium orthovanadate, 1 mM dithiothreitol, 0.25 mM ATP, and 37.5 mM MgCl₂ at 30 °C for the indicated times.

**Phospho-proteomic analysis**. The Kinoview™ and PhosphoScan™ assays were performed by Cell Signaling Technology Inc.

**Protein microarray binding assay**. The ProtoArray® assay, using the Human Protein Microarray platform, was performed by Invitrogen.

**High-performance molecular modeling platform**. Modeling and docking used the APPLIED Pipeline (Analysis Pipeline for Protein Ligand Interactions and Experimental Determination) at the Argonne Leadership Computing Facility, Argonne National Laboratory, tuned for the 786,432 core BlueGene/Q Mira[47], using a multi-stage pipeline that considers protein–protein/ligand interactions through evolutionary protein surface analysis[48–50], robust homology modeling[51],

massively parallel docking simulations using mixed strategies[52–56], and advanced, physics-based rescoring methodologies[56–58], all as described by us.

**LUMIER assay**. The LUMIER assay was performed as described, using reagents generously provided by that group[16].

**Statistical analysis**. All results were analyzed by a dedicated statistician (B.J.). Unless otherwise stated, statistical significance was evaluated with the two-sided Student's t-test using a threshold of $P < 0.05$. All experiments, unless otherwise stated, were conducted in replicates of at least N = 2–6 (with specific N values are denoted for each experiment) and were repeated at a separate time, also in replicates of N = 2–6. The relationship between dose and metastasis, and between drug concentration and effect on protein degradation, was evaluated by two-sided analysis of variance. The survival of mice was evaluated by the log-rank (Mantel-Cox) test.

**Data availability**. The data that support the findings of this study are available from the corresponding author upon reasonable request.

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

## Acknowledgements

Financial support for this research was provided to R.Ber. by the Veterans Administration (IBX002842A) and the National Institutes of Health (R01 CA122985, Prostate SPORE CA90386 and P30 CA069533), to J.B. by the National Institutes of Health (R01 GM086688) and to A.B. from the National Institutes of Health (GM094585), resources of the Argonne Leadership Computing Facility at Argonne National Laboratory (supported by the Office of Science of the US Department of Energy under contract DE-AC02-06CH11357) and allocations for computing (supported by the Department of Energy's Innovative and Novel Computational Impact on Theory and Experiment (INCITE) program). We wish to thank I. Ogden for technical support in performing cell and molecular-based assays, M. Lin for help in performing luciferase assays, A. Yemelyanov for help in transfection, and Mikko Taipale and Susan Lindquist for generously providing LUMIER reagents.

## Author contributions

Experiments were conceived and designed by L.X., R.G., A.P., A.B., M.A., S.K., A.M., W.A., K.S., and R.Ber. Experiments were performed by L.X., R.G., R.F., A.P., A.B., X.H., M.A., S.K., E.V., J.P., J.C., J.B., A.M., A.N., L.L., S.P., M.V., R.Bet., and G.F. Statistical analysis was performed by B.J. and R.Ber. The manuscript was written by L.X., R.G., R.F., and R.Ber.

## Additional information

**Competing interests:** R.Ber. and K.S. jointly hold a US patent for KBU2046 and jointly own Third Coast Therapeutics, and could thereby benefit financially from findings described herein. The remaining authors declare no competing interests.

