## [Peer Review File · Nature Communications]

Reviewers' comments:

Reviewer #1 (Remarks to the Author):

This manuscript by Xu et al. describes identification and validation of a novel flavonoid-based small molecule able to inhibit cell motility and invasion in vitro and experimental metastasis in vivo. Much of the data is compelling and would be of interest to readers of Nature Communications, but some experimental procedures require a more thorough description and some of the data are too cursory to support the authors' mechanistic conclusions. Finally, some of the authors' explanations of their data are hard to follow and do not appear to be internally consistent.

Major concerns:

1. Does the compound (herein abbreviated as '46') have any impact on MMPs or other matrix-destroying activities? Is its effect on invasion mediated only by inhibiting motility?
2. With respect to the experiment shown in Fig. 3, where drug treatment only prior to tumor cell implantation is effective, why did the authors wait for 3 days after implantation to begin treatment of the 'post-implantation' group? In this model, how long does it take to see metastasis to the jaw? Description of this experiment in both the figure legend and in Supplemental Methods is too cursory for the reader to understand how the treatment plan was selected. Would treatment with a higher dose of compound be more effective if initiated after implantation?
3. Of significance for addressing comment 2, what is the MTD with daily oral administration?
4. When was quantification of metastasis (see comment 2) performed? Again, this information is lacking in Supplemental Methods or in figure legends. What happens upon cessation of drug treatment – does metastasis eventually appear? If the drug does not directly impact cell viability, as the authors' in vitro data suggest, the expectation (at least for the prostate model) would be that cessation of drug treatment would eventually lead to some degree of metastasis. Did the authors look for this?
5. For the data in Fig. 4a, a comparable western blot showing levels of total Hsp90beta should be included to confirm that the reduced signal described in panel a really reflects reduced phosphorylation and not reduction in total Hsp90beta.
6. With respect to the wording used by the authors (bottom of 5th page of Results) to describe the impact of using S226A-Hsp90beta, I would argue that the data in Fig. 4c show that transfection of this mutant mimicked the effects of the drug, not 'abrogated' it, as written by the authors.
7. With respect to Figs. 4c – d, since there is no mention of knock down of endogenous Hsp90beta prior to transfection with WT, S226A or S226D mutants, why would transfection of either of the mutants abrogate the impact of the drug on the endogenous Hsp90beta, which presumably is still functional? Along these lines, expression levels of WT, S226A and S226D Hsp90beta mutants should be shown by western blot along with VC cells to allow the reader to visualize (a) the degree of over-expression of the Hsp90beta constructs, and (b) whether expression levels of WT, S226A and S226D transfected proteins are comparable to each other.
8. On the bottom of the last paragraph of 5th page of results (discussing the data in Fig. 4d), the authors state that cells transfected with S226D exhibited increased invasion but this is only true when comparison is made to vector control cells; cells transfected with WT Hsp90beta actually demonstrate more invasion than is seen in either VC cells or in cells transfected with S226D-Hsp90beta.
9. Top of next page (6th page of Results): the authors state that their data show that the post-translational modification of Hsp90beta can be induced by a small molecule. I would argue just the opposite – namely, that their data show that phosphorylation of Hsp90 can be inhibited by a small molecule.

10. In the same sentence, the authors write that this effect on Hsp90beta phosphorylation "is associated with selective inhibition of cancer cell motility" by their drug. "Associated with" suggests a mechanistic link between the two events and proof of such a link would require more experimentation. For this reason, I would strongly suggest replacing "is associated" with "may be associated".

11. I assume that casein kinase 2 (CK2) is among the panel of kinases examined by the authors in their Kinome Screen, but no direct mention is made of it. Since CK2 is known to phosphorylate both S226 in Hsp90beta and Cdc37, a simple explanation of the mechanism of this drug would be to inhibit CK2 activity. Have the authors specifically demonstrated that this is not the case? If they haven't, I strongly suggest that they do.

12. On top of the 8th page of the Results (where the authors discuss data in Fig. 5b-d and in Supplementary Fig. 18), they write that modeling suggests that Cdc37 binding to Hsp90 involves hydrogen bond formation with Glu33 of Hsp90beta to create formation of a new pocket into which the drug binds. This statement seems to be internally inconsistent with their later statement in the Discussion (middle of 2nd page) that their drug "did not bind to Hsp90". The authors need to clarify what they mean (e.g., did not bind to Hsp90 in the absence of Cdc37 binding?).

13. Lastly, the authors should be aware of a model of Cdc37 sampling of kinases and forming a tertiary complex with Hsp90, recently proposed by Agard and colleagues (<https://www.ncbi.nlm.nih.gov/pmc/articles/PMC5373496/>) that is distinct from the earlier model suggested by the Pearl group and derived from co-crystal structures. There is a possibility that both models may be correct and represent temporally distinct interaction poses of Cdc37, client and Hsp90, but the authors should at least consider their proposed binding model in the context of the Agard group's work and comment on its potential significance in that light.

Additional comment:

1. Figure 4c: if the data are expressed as %VC, should not the VC bar be set at 100 (as in Fig. 4d)?

Reviewer #3 (Remarks to the Author):

The notion of Precision Targeted Therapy of Tumor Cell Motility has been a long sought after goal of researchers and the authors present an interesting series of chemical modifications of genistein to derive a related compound termed KBU2046.

This drug is not highly cytotoxic to cells growing in 2D culture and has limited effects on primary tumor growth. They authors try to build support for a specific anti-metastasis effect via indirect effects on HSP90beta S226 phosphorylation. The data is weak. Primarily, 10 micromolar KBU2046 treatment of cells for 3 days was used to evaluate effects on cell movement (no dose-dependence shown and too long of a treatment to establish cause & effect). In vitro cell invasion values were presented as percent of control and maximal inhibition was approximately 50%. In mice, the authors claim that anti-metastasis effects occur at plasma concentrations of KBU2046 less than 100 nM. There is a disconnect with these values and the targets that they are evaluating (HSP90, BRaf, CDC37, RIPK1) are not necessarily motility-specific proteins. Missing are experiments to connect signaling changes occurring in vitro to what might be occurring within a primary tumor to prevent the local spread of tumor cells. There are many possible alternative explanations. I do not find this study compelling.

Response to Reviewers' comments:

We wish to thank the reviewers for their consideration of our manuscript. Their review and critiques have served to improve this manuscript. Below is a point-by-point response to reviewer critiques. Changes in the manuscript and associated documents are underlined.

Reviewer #1:

General comments: Much of the data is compelling and would be of interest to readers of Nature Communications, but some experimental procedures require a more thorough description and some of the data are too cursory to support the authors' mechanistic conclusions...

Response. Thank you for your comprehensive and informed review, we are encouraged that you felt that our data would be compelling to the readers of Nature Communications. In our revised manuscript as well as below, you will find that we have provided the requested clarifications, expanded descriptions and additional experiments requested.

1. Does the compound (herein abbreviated as '46') have any impact on MMPs or other matrix-destroying activities? Is its effect on invasion mediated only by inhibiting motility?

Response. This is an excellent and logical question. We did in fact suspect that this may be contributing to KBU2046's anti-invasion efficacy, and as such we evaluated an entire panel of MMPs as part of two separate expression arrays. The Human Extracellular Matrix and Adhesion Array and The Human Metastasis and Invasion Array, both from Qiagen, and together they included the following MMPs: 1,2,3, and 7-16, as components of their panels. However, no significant differences in MMPs or other members of the arrays were observed. This finding is consistent with our identification of RAF1 as an important mediator of KBU2046 efficacy, as RAF1 has been shown to modulate motility through direct effects on the cytoskeleton. As the Qiagen array data is all negative, we have elected not to include it, but would be happy to if requested. The general experimental design involved treatment of human prostate cancer PC3-M cells for three days with KBU2046, measuring gene expression per manufacture's instructions, and conducting experiments N=2 separate times. This was done for each array; i.e., N=4 experiments total.

2. With respect to the experiment shown in Fig. 3, where drug treatment only prior to tumor cell implantation is effective, why did the authors wait for 3 days after implantation to begin treatment of the 'post-implantation' group? In this model, how long does it take to see metastasis to the jaw? Description of this experiment in both the figure legend and in Supplemental Methods is too cursory for the reader to understand how the treatment plan was selected. Would treatment with a higher dose of compound be more effective if initiated after implantation?

Response.

- i. Drug treatment prior to implantation was effective, as expected, reflecting its ability to inhibit the cell motility required to invade into a distant metastatic site. Drug treatment beginning 3 days post implantation was ineffective, as expected, reflecting the fact that cells have already completed the necessary motility steps for establishing distant metastasis.
- ii. Presence of cells within the jaw can be observed (by IVIS imaging) within 30 minutes after implantation. Weekly IVIS imaging thereafter demonstrates persistence and increased growth.
- iii. Experimental details have now been expanded. See legend for Fig 3 and supplemental methods, section entitled "Animal Models of Human Cancer Metastasis and of Systemic Effects" section.

iv. Regarding higher doses of drug, a dose-response relationship was examined in the orthotopic implantation model (Figs 2a and b). Those findings demonstrated increased efficacy with increased dose. However, in the post-implantation model we would not anticipate increased efficacy with higher doses due to fact that tumors are already established and because KBU2046 does not directly inhibit tumor growth.

3. Of significance for addressing comment 2, what is the MTD with daily oral administration?

Response. We have tested oral doses up to 150 mg/kg with no observed toxicity (Fig 2a and b). Those studies were comprehensive, involving treatment for up to 35 days, and histologic examination of all critical organs (Supplementary Figs. 7-8). Based upon in vitro studies, inclusive of 14 day human stem cell hematopoietic colony formation assays, wherein stem cells (which tend to be very sensitive to drug toxicity) were exposed to up to 40 micromolar of drug for 14 days, and no toxicity was observed (Fig. 1e), we expect that the MTD after oral administration in animals to be much higher. We had previously described in supplemental data (Supplementary Figure 8) that there was no observed toxicity in 14 day toxicity studies after intravenous (IV) injection of 15, 75 or 125 mg/kg-body weight of KBU2046. We now add that IV injection of 150 mg/kg led to immediate death.

4. When was quantification of metastasis (see comment 2) performed? Again, this information is lacking in Supplemental Methods or in figure legends. What happens upon cessation of drug treatment – does metastasis eventually appear? If the drug does not directly impact cell viability, as the authors' in vitro data suggest, the expectation (at least for the prostate model) would be that cessation of drug treatment would eventually lead to some degree of metastasis. Did the authors look for this?

Response.

i. Quantification was performed once a week for 4 weeks starting 7 days post implantation as indicated in Figure 3a. This has been clarified in the legend of fig 3 and in the supplemental methods section, entitled "Animal Models of Human Cancer Metastasis and of Systemic Effects".

ii. Regarding the cessation of treatment, we did test this, and the reviewer is correct. Specifically, as shown in Figs 3b and c, the maximum therapeutic benefit was observed in mice receiving continuous therapy (i.e., from 3 days pre-IC injection through the full 4 weeks of the experiment); this cohort of mice was designated the "Pre" cohort. However, in the "Pre7Stop" cohort, mice received initial treatment identical to the Pre cohort (beginning 3 days pre-IC injection), but it was then stopped after day 7 post IC injection, i.e. no treatment for the last 3 weeks of the experiment. This Pre7Stop cohort of mice experienced a significant decrease in total body metastasis compared to that of control mice. However, our findings also suggested that the therapeutic benefit observed in the Pre7Stop cohort was not as strong as that when treatment was continuous, as was the case in the Pre cohort. This is supported by metastasis to the jaw, for which this model is designed, which in the Pre7Stop cohort are intermediate in value between that of Pre cohort mice (receiving maximum therapy) and control mice. Note that the Post cohort of mice started receiving treatment 3 days after intra-cardiac injection, did not demonstrate any therapeutic benefit, and was identical to control mice. We have clarified our description of these data in the Results section of the manuscript, in association with Fig 3.

5. For the data in Fig. 4a, a comparable western blot showing levels of total Hsp90beta should be included to confirm that the reduced signal described in panel a really reflects reduced phosphorylation and not reduction in total Hsp90beta.

Response. This figure is now included as part of the supplemental figure 14b. Note, no change in total HSP90 β levels were observed.

6. With respect to the wording used by the authors (bottom of 5th page of Results) to describe the impact of using S226A-Hsp90beta, I would argue that the data in Fig. 4c show that transfection of this mutant mimicked the effects of the drug, not 'abrogated' it, as written by the authors.

Response. We agree completely. This non-phosphorylatable mutant exactly mimicked the effect of the drug. As drug decreased phosphorylation, this was an expected finding in the non-phosphorylatable mutant. In the face of this mutant, further effects of the drug were abrogated. This too was expected, as we have effectively removed the site upon which the drug acts. We have clarified this in the revised manuscript; see Results, Fig 4c.

7. With respect to Figs. 4c – d, since there is no mention of knock down of endogenous Hsp90beta prior to transfection with WT, S226A or S226D mutants, why would transfection of either of the mutants abrogate the impact of the drug on the endogenous Hsp90beta, which presumably is still functional? Along these lines, expression levels of WT, S226A and S226D Hsp90beta mutants should be shown by western blot along with VC cells to allow the reader to visualize (a) the degree of over-expression of the Hsp90beta constructs, and (b) whether expression levels of WT, S226A and S226D transfected proteins are comparable to each other.

Response. The expression levels of the transfected proteins are now included in supplementary data fig 15, and are comparable. Further, while cells do retain native HSP90 β , as is shown in supplementary fig 15, transfected mutants dominate. Independent knockdown experiments yield similar and expected findings (supplementary fig 15).

8. On the bottom of the last paragraph of 5th page of results (discussing the data in Fig. 4d), the authors state that cells transfected with S226D exhibited increased invasion but this is only true when comparison is made to vector control cells; cells transfected with WT Hsp90beta actually demonstrate more invasion than is seen in either VC cells or in cells transfected with S226D-Hsp90beta.

Response. This is correct. Cells transfected with wild type (WT) HSP90 β were most invasive, followed by cells transfected with pseudophosphorylated mutant S226D-HSP90 β , which were in turn more invasive than vector control (VC) cells. This was not an unexpected finding. Pseudophosphorylated mutants, such as S226D, mimic phosphorylated residues and thus render a degree of constitutive activity. However, they are in fact altered proteins and rarely function as efficiently as unaltered wild type. Therefore, over expressing a wild type provides more “function” than over expressing a mutant protein. Therefore, it was not unexpected that invasion followed the pattern: WT HSP90 β > S226D-HSP90 β > VC. It should be noted that the central point of this experiment was to assess drug efficacy in the face of constitutive activity. By mutating the site to an active/functional state, a drug cannot alter it, and therefore the drug loses efficacy. This is exactly what we demonstrate. Specifically, drug activity is retained in VC and WT cells, and is essentially lost in S226D cells, where its effect is in fact not statistically significant. We now provide an expanded description of these elements; see Results, Figs 4c and d.

9. Top of next page (6th page of Results): the authors state that their data show that the post-translational modification of Hsp90beta can be induced by a small molecule. I would argue just the opposite – namely, that their data show that phosphorylation of Hsp90 can be inhibited by a small molecule.

Response. We agree. Our intent was to convey that our small molecule was affecting a post translation modification. We have clarified this point in the revised manuscript; see Results, Fig 4d.

10. In the same sentence, the authors write that this effect on Hsp90beta phosphorylation “is associated with selective inhibition of cancer cell motility” by their drug. “Associated with” suggests a mechanistic link between the two events and proof of such a link would require more experimentation. For this reason, I would strongly suggest replacing “is associated” with “may be associated”.

Response. The wording has been clarified in the revised manuscript; see Results, Fig 4d.

11. I assume that casein kinase 2 (CK2) is among the panel of kinases examined by the authors in their Kinome Screen, but no direct mention is made of it. Since CK2 is known to phosphorylate both S226 in Hsp90beta and Cdc37, a simple explanation of the mechanism of this drug would be to inhibit

CK2 activity. Have the authors specifically demonstrated that this is not the case? If they haven't, I strongly suggest that they do.

Response. This is an excellent point, we have done this, and have demonstrated that CK2 is not affected by KBU2046. Specifically, because of CK2's known link to S226 we likewise suspected that it could be involved in the mechanism of action. CK2 was indeed included in the LUMIER assay (see Fig. 6 and Supplementary Fig. 19). We used the LUMIER assay to evaluate whether KBU2046 was altering the binding of client protein kinases to HSP90 β /CDC37 heterocomplexes in intact cells, and thereby evaluated hundreds of client protein kinases. In this manner, we observed that KBU2046 did not alter the binding of CK2 to the HSP90 β /CDC37 heterocomplex in cells. However, because of the known mechanistic link, we also utilized our in vitro system to interrogate purified CK2 and specifically found that KBU2046 does not inhibit CK2's ability to phosphorylate S226 on HSP90 β . This is shown below, and if requested, we are happy to insert these otherwise negative findings into supplementary data.

CK2 in vitro kinase assay. As described in Figs 6f and g, the ability of recombinant CK2 to phosphorylate serine 226 on HSP90 β , in the presence of CDC37, was evaluated +/- KBU2046 (46). Reaction time in minutes (Mins) is denoted. KBU2046 had no effect upon serine 226 phosphorylation.

12. On top of the 8th page of the Results (where the authors discuss data in Fig. 5b-d and in Supplementary Fig. 18), they write that modeling suggests that Cdc37 binding to Hsp90 involves hydrogen bond formation with Glu33 of Hsp90beta to create formation of a new pocket into which the drug binds. This statement seems to be internally inconsistent with their later statement in the Discussion (middle of 2nd page) that their drug "did not bind to Hsp90". The authors need to clarify what they mean (e.g., did not bind to Hsp90 in the absence of Cdc37 binding?).

Response. The drug only binds to hsp90 and cdc37 when both proteins are present, it does so by binding in a pocket created at the interface of the two proteins. When either protein is present alone, there is no binding. We now clarify this point in the revised manuscript, and do so in Results when discussing Fig 5b-d and in the Discussion (second last paragraph).

13. Lastly, the authors should be aware of a model of Cdc37 sampling of kinases and forming a tertiary complex with Hsp90, recently proposed by Agard and colleagues (<https://www.ncbi.nlm.nih.gov/pmc/articles/PMC5373496/>) that is distinct from the earlier model suggested by the Pearl group and derived from co-crystal structures. There is a possibility that both models may be correct and represent temporally distinct interaction poses of Cdc37, client and Hsp90, but the authors should at least consider their proposed binding model in the context of the Agard group's work and comment on its potential significance in that light.

Response. We are aware of these findings, agree completely with this important point, and have included a discussion of this in a revised manuscript; see second last paragraph of Discussion.

Additional comment 1. Figure 4c: if the data are expressed as %VC, should not the VC bar be set at 100 (as in Fig. 4d)?

Response. The figure has been changed as suggested.

Reviewer #3:

1. This drug is not highly cytotoxic to cells growing in 2D culture and has limited effects on primary tumor growth.

Response. Exactly. Its specific action is upon inhibiting motility. Which was our goal, has been a long sought after goal, and is of critical importance to cancer biology and therapeutics.

2. They authors try to build support for a specific anti-metastasis effect via indirect effects on HSP90beta S226 phosphorylation. The data is weak. Primarily, 10 micromolar KBU2046 treatment of cells for 3 days was used to evaluate effects on cell movement (no dose-dependence shown and too long of a treatment to establish cause & effect).

Response. It is not logical to stipulate that 3 days treatment of cells is too long. We demonstrated the role of S226 through point mutation studies, by demonstrating effects in tumors of treated animals, and we demonstrated it in an in vitro system of only recombinant purified proteins, in an in vitro kinase assay, after 30 minutes of treatment (fig 6f). Further, in fig 2, we demonstrate comprehensive assessment of dose, do so in animals, demonstrate efficacy at low nanomolar plasma concentrations, and then go on to demonstrate efficacy in a total of three different animal model systems (figs 2 and 3).

2. In vitro cell invasion values were presented as percent of control and maximal inhibition was approximately 50%.

Response. This is correct. It is a routine way of reporting such, by us and others. After 3 days treatment of cells, effect upon invasion is 50%. After longer treatment of animals, effect is >90%. This is exactly what is expected, and represents the effect of treatment time on efficacy.

3. In mice, the authors claim that anti-metastasis effects occur at plasma concentrations of KBU2046 less than 100 nM.

Response. Figs 2a and b definitively demonstrate >90% suppression of metastasis at a plasma concentration of 24 nM.

4. There is a disconnect with these values and the targets that they are evaluating (HSP90, BRaf, CDC37, RIPK1) are not necessarily motility-specific proteins. Missing are experiments to connect signaling changes occurring in vitro to what might be occurring within a primary tumor to prevent the local spread of tumor cells. There are many possible alternative explanations. I do not find this study compelling.

Response. The reviewer is confusing Braf with RAF1 (RAF1 is also known as CRaf). RAF1 has well established links to motility. Further, we linked effects between HSP90/CDC37, and the client proteins, RAF1 and RIPK1, using the LUMIER assay. The LUMIER assay is well established and recognized as a gold standard for evaluating the interaction between HSP90 and its clients (Taipale et al, 2012 Cell). Because of RAF1's well established links to motility, we focused studies on it. We then did in fact conduct in vitro cell-based studies that definitively demonstrated that RAF1 regulated cell motility in our system and that its presence was necessary for drug action (Fig 6d). We also conducted studies involving purified recombinant RAF1, demonstrating KBU2046-mediated inactivation (Fig 6e).

Reviewers' comments:

Reviewer #1 (Remarks to the Author):

The authors have thoroughly and convincingly addressed the points raised by the reviewers. Their revised manuscript contains new experiments and other clarifications, and is a much stronger manuscript. All of my concerns have been addressed.

Reviewer #3 (Remarks to the Author):

The manuscript entitled "Precision Therapeutic Targeting of Human Cancer Cell Motility" by Xu, et. al., describes a strategy used to determine how their lead compound, KBU2046, appears to induce anti-metastatic activity. While at first glance this article appeared intriguing, but after looking in more detail, too many questions and concerns exist to support the enthusiasm needed for publication.

First of all, the results section is written in a very colloquial manner, rather than matter of fact. Secondly, the lead compound is ketone that is likely to react with many different proteins, most likely through formation of a Schiff base (confirmation through hydride trapping may prove useful). Thirdly, the scaffold appears to be promiscuous in nature, its not clear why the PI evaluated this for kinase inhibition, as it does not exhibit such a pharmacophore. Since Hsp90B is an essential protein, it is not clear how this molecule exhibits a non-toxic and proliferative activity- does not reflect the proposed paradigm. The PI states that the compound does not bind CDC37 or Hsp90B directly, but the DARTS assay shows protection of CDC37 and degradation of Hsp90B- clearly demonstrating that the molecule binds CDC37-Should be amenable to determination, perhaps via trapping of the imine/Schiff base (which will also hit other targets). Previous studies have shown that celastrol also disrupts interactions between CDC37 and Hsp90, it has also been proposed to do so in a covalent manner. Note that CDC37 bind to both Hsp90a and Hsp90b, which further complicates the proposed model. The Docking model utilized in these studies is very speculative, the Agard structure would be most useful for comparison. If the molecule did bind and exhibit the proposed activity, then it remains unclear as to why it inhibits phosphorylation of the Ser226, and no other (other substrates as well).

Significance of pretreating animals with molecule before tumor implantation seems to lack parallel to normal cancer progression. Concentrations are missing from most of the Figures for both "controls" and lead compound.

While there are a lot of studies reported in this manuscript, the lead compound, its reactive properties, and what appears to be contradictory mechanisms of action dampen enthusiasm for this manuscript.

Response to Reviewers' comments:

Reviewer #1:

The authors have thoroughly and convincingly addressed the points raised by the reviewers. Their revised manuscript contains new experiments and other clarifications, and is a much stronger manuscript. All of my concerns have been addressed.

Response. We are very appreciative of this reviewer's assessment.

Reviewer #3:

1. The results section is written in a very colloquial manner, rather than matter of fact.

Response. We had sought to match the tone of the manuscript to papers published in *Nature Communications*, are happy to alter it, and will seek advice in this regard from the editor.

2. The lead compound is ketone that is likely to react with many different proteins, most likely through formation of a Schiff base (confirmation through hydride trapping may prove useful).

Response. This point, and others below, relate to chemical structure, and address what we consider to be very important aspects of our work. We would like to highlight that one of the authors, Karl Scheidt, is a Professor of Chemistry, Northwestern University, and guided our synthetic strategy cognizant of the impact of chemical structure on reactivity, and ultimately activity and specificity. It is unlikely that KBU2046 reacts irreversibly with any biological nucleophiles (proteins) since any Schiff base-forming process would be reversible. In fact, there are multiple FDA approved drugs that have ketones, not esters or amides. These include, but are not limited to: nambumetone, raloxifene, oxycondone, normethadone, eprazinone, and multiple steroids (tibolone, prednisone).

This statement is further supported that across a wide array of experiments in this manuscript there was a lack of toxicity with KBU2046. If KBU2046 were promiscuous and undergoing widespread protein interactions, then such high tolerability, i.e., low toxicity, is highly unlikely. It is important to note that the nature of the investigations we conducted in this regard were very comprehensive. They involve: i. multiple molecular level screening assays, including proteomic-based profiling of whole cell lysates (Supplementary fig. 11), screens for inhibition of kinase activity, across three separate screening platforms (Supplementary fig. 13), screens for estrogenic activity (Fig. 1f and Supplementary fig. 4), and pathway focused screens for effects on pathways previously shown to play a role in regulating human prostate cell motility (Supplementary fig. 10); ii. a series of dedicated cellular level assays for toxicity, including growth inhibition assays across a panel of human prostate cell lines (Fig. 1d), screening for bone marrow toxicity using human cord blood stem cells to perform 14 day tri-lineage hematopoietic colony formation assays (Fig. 1e) screening across the NCI 60 panel of cell lines (Supplementary fig. 3) and lack of effects on the growth of human tumors in mice (Supplementary fig. 7); iii. comprehensive systemic screens for toxicity, including dedicated 35 day high dose treatment of mice followed by histological analysis of all critical organs (Supplementary fig. 8), dedicated 14 day murine-based toxicity studies followed by comprehensive clinical chemistry profiling and inclusive of both oral and iv dosing (Supplementary fig. 8), and across a total of five separate murine based efficacy and dedicated toxicity experiments conducted throughout this manuscript, there were no outward effects of toxicity (e.g., behavior change, weight loss, of any other) were observed. The only exception was immediate death after iv injection of 150 mg/kg KBU2046, which is a massive dose and associated osmotic load, and should not be considered otherwise informative.

2. The scaffold appears to be promiscuous in nature, its not clear why the PI evaluated this for kinase inhibition, as it does not exhibit such a pharmacophore.

Response. These are important points. With respect to the chemical scaffold, isoflavones and isoflavanones are actually well established and recognized “privileged” scaffolds for biological activities based on a survey of the literature. These compounds display a broad array of interesting effects and was a logical starting point for medicinal chemistry development geared on generating more selective compounds. We have achieved this goal by chemically editing this known class of compounds to elicit a desirable phenotype (i.e., anti-motility) and then elucidated the molecular mechanism.

With respect to why we pursued kinase inhibition studies, there were three reasons that pointed us in that direction. First, genistein was our starting compound, and is known to inhibit kinases. Second, our phosphoproteomics screen demonstrated decreased phosphorylation of serine 226 on HSP90beta, consistent with kinase inhibition. Third, our prior studies had demonstrated that genistein inhibited prostate cell motility by inhibiting the kinase MEK4/MKK4 (at low nanomolar concentrations, i.e., three logs lower than that associated with promiscuous inhibition of a wide array of kinases). Based upon this data, we believed that KBU2046 was acting by directly inhibiting kinase activity, and exhaustively pursued this mechanism of action. After thoroughly confirming that this was not correct, we went on to demonstrate that KBU2046 did in fact exert its effect through inhibition of Raf1 kinase, but it did so indirectly, through modulating HSP90beta/CDC37-mediated modulation of Raf1 activation (Figs. 6a-c).

3. Since Hsp90B is an essential protein, it is not clear how this molecule exhibits a non-toxic and proliferative activity-does not reflect the proposed paradigm. The PI states that the compound does not bind CDC37 or Hsp90B directly, but the DARTS assay shows protection of CDC37 and degradation of Hsp90B-clearly demonstrating that the molecule binds CDC37-Should be amenable to determination, perhaps via trapping of the imine/Schiff base (which will also hit other targets).

Response. We now more clearly state that KBU2046 binds to HSP90beta-CDC37 heterocomplexes, but will not bind HSP90beta or CDC37 proteins when they do not co-exist in a heterocomplex. As stated by the reviewer, KBU2046 does provide protein protection in a concentration-dependent manner in a DARTS assay only when heterocomplexes are present (Fig. 5a). However, when the same DARTS assay conditions are conducted using only HSP90beta or CDC37 alone, there is no protection (Supplementary fig. 16). See Results page 11, paragraph 1, sentence 6 (5th new sentence) and Discussion page 15, last paragraph, sentences 3-4.

We now more clearly describe how our data reflects our proposed paradigm. There are major functional differences between classic HSP90 inhibitors and KBU2046. Classic HSP90 inhibitors physically bind directly to HSP90 when only HSP90 is present, alter binding and stability of up to hundreds of client proteins, and are potent inducers of cell toxicity and death. In contrast, KBU2046 only binds to heterocomplexes, and exhaustive studies inclusive of DARTS, isothermal titration calorimetry, fluorescent thermal shift assay, and bilayer interferometry all fail to indicate that KBU2046 will bind HSP90 in the absence of CDC37 (Fig. 5a and Supplementary Fig. 16). Further our LUMIER assay demonstrates that KBU2046 only effects binding of 1% of client proteins to the HSP90beta/CDC37 complex, selectively effecting only those that regulate cell motility, and does not cause widespread degradation of client proteins (Fig. 6a and Supplementary fig. 19a). Finally, KBU2046 is non-toxic. These functional differences are accompanied by differences in binding. Our 3D structural model places KBU2046 in a cleft formed at the interface of HSP90beta and CDC37 binding. While our structural model is in silico, it was also constructed based upon physical studies, inclusive of chemical cross-linker based approaches to defining structure, DARTS findings and x-ray crystallographic based structural analysis.

Our proposed paradigm comes from and is directly supported by these findings. KBU2046 binds to a cleft present in heterocomplexes (DARTS, 3D model), this has a precision-type effect upon chaperone function resulting in small but selective effects on client proteins that regulate motility (supported by cell based LUMIER assays), resulting in selective inhibition of cell motility through decreased activation of Raf1 (Fig. 6c-e) and lack of toxicity. Please see Results page 11, paragraph 1, last 3 sentences, page 12, last paragraph, second last sentence and Discussion, page 16, last paragraph.

4. Previous studies have shown that celastrol also disrupts interactions between CDC37 and Hsp90, it has also been proposed to do so in a covalent manner.

Response. By comparing the chemical structure of KBU2046 to that of celastrol, it is evident that there are many structural differences between these two compounds (see below). Importantly, celastrol is well known and highly activated “quinone methide” known to undergo covalent interaction with proteins (thiol addition), see *J. Org. Chem.* **1965**, 30, 1729 and *J. Am. Chem. Soc.* **2011**, 133, 19634.

Consistent with these differences in chemical structure and reactivity, celastrol has been shown to exert a relatively wide array of effects, including TRIAL-mediated induction of apoptosis (*J. Biol. Chem.* **2010**, 285, 11498), angiogenesis through inhibition of VEGF-mediated activation of the AKT/mTOR/P70S6K pathway (*Cancer Research*, **2010**, 70, 1951), anti-malarial action through inhibition of the enzyme, enoyl-acyl carrier protein reductase (*Bioorg Med Chem.* **2014**, 22, 6053), inhibition of lung cancer cell growth through inhibition of PP2A (*Carcinogenesis*, **2014**, 35, 905), effects upon lung cancer through proteasomal degradation (*Cancer Sci.* **2015**, 106, 902) and many others.

5. Note that CDC37 bind to both Hsp90a and Hsp90b, which further complicates the proposed model.

Response. Several findings by us demonstrate that HSP90beta is the important pharmacologic target. We used a LUMIER assay to screen for changes in client protein binding induced by KBU2046 in intact cells (Fig. 6a and Supplementary fig. 19a). This assay used HSP90beta exclusively, and was shown by the developers of this assay (published in *Cell*, 2012) to measure client protein binding to HSP90beta/CDC37 heterocomplexes. Hits were then investigated with *in vitro* assays using only purified proteins, HSP90beta, CDC37, Raf1, thereby demonstrating a relevant pharmacologic effect of KBU2046 (Fig. 6e). In corroborating studies, we constructed point mutants of HSP90beta, demonstrating that it is necessary and sufficient (depending upon mutation status) for affecting KBU2046 efficacy (Fig. 4b-d). Complementing these latter studies, we conducted HSP90beta specific knockdown studies (which we showed did not knockdown HSP90alpha), which provided additional confirmation that HSP90beta is necessary for KBU2046 efficacy (Supplementary fig. 15).

6. The Docking model utilized in these studies is very speculative, the Agard structure would be most useful for comparison.

Response. It is well recognized that there are several existing models, including an earlier model suggested by the Pearl group derived from co-crystal structures, the more recent model proposed by the Agard group and the one we propose. We do want to emphasize that our model, although *in silico*, is informed by experimental data, inclusive of chemical cross-linking studies designed to probe physical structure, x-ray crystallographic structure and findings from our DARTS experiments (Fig. 5 and Supplementary Fig. 18). We would also like to highlight that one of the authors, Andrew Binkowski, is an expert in computer-aided analysis of protein structure, is a scientist at Argonne National Laboratory and the University of Chicago, and as detailed in Supplemental fig. 18 used state-of-the-art methods and facilities to construct our proposed model, including the multi-stage APPLIED Pipeline (Analysis Pipeline for Protein Ligand Interactions and Experimental Determination) at the Argonne Leadership Computing Facility, run on the 786,432 core BlueGene/Q Mira supercomputer, and deploying a suite of associated novel tools developed on-site and previously reported in the literature.

Importantly the existence of one model does not preclude the possibility of the other models being correct, each may represent temporally distinct interaction poses of Cdc37, client and Hsp90. In recognition of these very important points we have provided an expanded discussion of this topic, emphasizing the proposed nature of our model and the need for deeper investigations. Please see Discussion, page 16, paragraph 2, sentences 2-3 and the last sentence.

7. If the molecule did bind and exhibit the proposed activity, then it remains unclear as to why it inhibits phosphorylation of the Ser226, and no other (other substrates as well).

Response. It is likely that there are additional effects, but that they are below our level of detection and their functional significance is below that of effects on ser226. Specifically, we probed for global changes in protein phosphorylation, and only identified a change in the phosphorylation state of ser226 (Fig. 4a and Supplementary figs. 11 and 14). It is likely there are other changes, but they are below the sensitivity of this technique. However, as described in detail above, we have conducted a series of experiments to probe the functional relevance of ser226, and have demonstrated it.

Our proposed model is consistent across experimental data and proposed structure. KBU2046 binds to a site that is only created when HSP90beta and CDC37 are together in a heterocomplex, this results in a small yet precise change in chaperone function that selectively alters the binding and activation profile of a select few client proteins that regulate cell motility. Recognizing the complexity of the system and the dynamic nature of the client-chaperone complex we suspected that KBU2046-mediated inhibition of Ser226 phosphorylation was not an isolated event (i.e. not mediated by a single kinase operating in isolation). We explored this possibility in our *in vitro* system by investigating the interplay of multiple kinases that we previously detected in our LUMIER assay. Specifically, in intact cells KBU2046 increases MAP3K6 binding to complexes (Fig. 6a). However, MAP3K6 is not predicted to phosphorylate the Ser226 motif. We then went on to demonstrate that when MAP3K6 is added to the *in vitro* kinase assay system, SGK3-mediated phosphorylation of HSP90β is not only inhibited, but in fact decreases in the presence of KBU2046 (Fig. 6g), emulating what is seen in intact cells.

8. Significance of pretreating animals with molecule before tumor implantation seems to lack parallel to normal cancer progression. Concentrations are missing from most of the Figures for both "controls" and lead compound.

Response. Our models included both pre- and post-treatment cohorts, and were designed to probe various aspects of cell motility in the context of the metastatic cascade. For example, in the prostate orthotopic implantation model, pre-treatment of mice ensured that drug-mediated

efficacy was present from the moment of surgical implantation (Fig. 2a). This was important as our goal was to assess movement of cells out of the primary organ, and we thereby demonstrated that KBU2046 did inhibit this. Of note, associated blood concentrations are displayed in Fig. 2b. However, in our cardiac injection model, we were seeking to address drug efficacy upon latter steps of the metastatic cascade, i.e. when cells have already left the primary organ (Fig. 3). In this model, we built in pre-treatment and 3 day post-treatment cohorts. Pre-treatment addressed the question if KBU2046 would inhibit distal implantation; the answer was yes, and was an anticipated finding. Post-treatment addressed the question if KBU2046 would inhibit development of established metastasis and resultant end organ destruction (in this case bone) after cells had already implanted; the answer was no, and was an anticipated finding.

With respect to concentrations, for animal studies, we measured blood concentrations for prostate orthotopic implantation studies in male mice and depicted them in Fig. 2b. For comprehensive pharmacokinetic studies, conducted on female mice, they are depicted in Fig. 2c and Supplementary fig. 6. Due to the comprehensive and concordant nature of these studies, we did not further measure blood concentrations in other animal studies. For all Figures and all animal studies, doses are listed. With respect to cell culture based studies, concentration data were provided for all studies, was presented in the Figures or legends for initial studies, and was provided in supplementary data for experiments in Figures 4 and 6. We now provide this information in the legends for these Figures.

9. While there are a lot of studies reported in this manuscript, the lead compound, its reactive properties, and what appears to be contradictory mechanisms of action dampen enthusiasm for this manuscript.

Response. Our responses to specific queries above demonstrate that KBU2046 does not possess non-specific chemical reactivity, and that our findings support a unified mechanism of action, all supported by a comprehensive set of investigations.

Thank you in advance for considering our responses. We believe that we have addressed all current queries, all prior queries, and that this manuscript will be of high interest to readers of Nature Communications.

Reviewers' comments:

Reviewer #4 (Remarks to the Author):

Precision Therapeutic Targeting of Human Cancer Cell motility

There are still two serious issues with the manuscript's contents:

1. While Schiff-base formation may be reversible, the reverse reaction may not occur during the life of the protein. Considering KBU2046's binding affinity is proposed to be driven by a single H-bonding interaction with the carbonyl in question, studies should be conducted to ensure the formation of a Schiff base is not occurring.
2. The authors address many of the concerns raised by reviewer 3, and the clarified mechanism of action seems logical and supported by the results. However, considering the structure and proposed binding mode of KBU2046, it is incredibly difficult to believe that KBU2046 is selective for a single cleft on the surface of Hsp90 β (upon Cdc37 binding) and does not have other interactions.

While the authors approach the development of KBU2046 correctly (phenotypic screening, deselection for unfavorable PK/PD properties), selective-affinity driven by a single H-bonding interaction and a 'lipophilic interaction' seems far-fetched. Have the author's considered the thermodynamic profile of this compound? Binding is likely to be driven by desolvation which dampens the possibility KBU2046 is a selective inhibitor.

The authors do not (and likely cannot) rationalize the selectivity of KBU2046 to Hsp90 β , which reduces the impact of this manuscript.

Minorly:

3. The author's edited description of classic Hsp90 inhibitors is misrepresentative of classic Hsp90 inhibitors and the Hsp90 chaperone cycle. Classic Hsp90 inhibitors exhibit 200-fold selectivity for the Hsp90 heteroprotein complex over the Hsp90 homodimer. Classic Hsp90 inhibitors DO exhibit harsh on-target toxicities and new therapeutic avenues ARE needed, however, the novelty of KBU2046 is overstated.

Response to Reviewers' comments:

We wish to thank reviewers for their time and thoughtful review. Please find below point-by-point responses to reviewer queries. Associated changes in the manuscript are underlined.

Reviewer (your email indicates #3, the review letter indicates #4):

1. While Schiff-base formation may be reversible, the reverse reaction may not occur during the life of the protein. Considering KBU2046's binding affinity is proposed to be driven by a single H-bonding interaction with the carbonyl in question, studies should be conducted to ensure the formation of a Schiff base is not occurring.

Response. To specifically examine Schiff-base formation, we incubated KBU2046 with L-alanine at 27°C (18 h) and then 37°C for a total 72 hours, and observed no change in 1H or 13C NMR spectra of drug from the original observed spectra for KBU2046 alone. This exposure experiment demonstrates that there is no accumulation of the Schiff-base imine/enamine product in the presence of a biologically relevant, representative primary amine. These studies were conducted by co-author Karl Scheidt, Professor of chemistry at Northwestern University. Attached to this letter is a detailed description of the experiment.

These new experimental findings are concordant with our other findings presented in the manuscript. Specifically, if KBU2046 were forming significant amounts of long lasting Schiff-bases with proteins, or other biological nucleophiles, then significant damage to biological systems should result. We have conducted a highly comprehensive series of investigations, inclusive of multiple screens at the molecular level, the cellular level and the systemic level. Together, those findings demonstrate highly selective biological action and fail to identify off target toxicity.

2. The authors address many of the concerns raised by reviewer 3, and the clarified mechanism of action seems logical and supported by the results. However, considering the structure and proposed binding mode of KBU2046, it is incredibly difficult to believe that KBU2046 is selective for a single cleft on the surface of Hsp90 β (upon Cdc37 binding) and does not have other interactions.

While the authors approach the development of KBU2046 correctly (phenotypic screening, deselection for unfavorable PK/PD properties), selective-affinity driven by a single H-bonding interaction and a 'lipophilic interaction' seems far-fetched. Have the author's considered the thermodynamic profile of this compound? Binding is likely to be driven by desolvation which dampens the possibility KBU2046 is a selective inhibitor.

The authors do not (and likely cannot) rationalize the selectivity of KBU2046 to Hsp90 β , which reduces the impact of this manuscript.

Response. We agree, it is not possible for us to state that KBU2046 is not binding to multiple sites. This is true for any compound. Further, we agree that our 3D structure model is only that,

have now toned down the meaning of this model, and further highlight its speculative nature. Please see Discussion page 16, first new paragraph, sentence 2 and last two sentences.

With respect to biological selectivity, we have demonstrated inhibition of cell motility and metastasis across several model systems, and were not able to find evidence of off-target effects in investigations ranging from molecular, cellular and systemic. Based upon the exhaustive nature of these investigations, we feel they well justify our stating the biological selectivity of KBU2046.

3. The author's edited description of classic Hsp90 inhibitors is misrepresentative of classic Hsp90 inhibitors and the Hsp90 chaperone cycle. Classic Hsp90 inhibitors exhibit 200-fold selectivity for the Hsp90 heteroprotein complex over the Hsp90 homodimer. Classic Hsp90 inhibitors DO exhibit harsh on-target toxicities and new therapeutic avenues ARE needed, however, the novelty of KBU2046 is overstated.

Response. We would like to respectfully point out that we never made any such claims that classic inhibitors did not have increased affinity to binding Hsp90 in heterocomplexes. We did state that they are recognized for their ability to bind Hsp90 in the absence of other proteins. And we did highlight the harsh/toxic effects of classic inhibitors.

Experimental

Workflow for sample prep and analysis

¹H and ¹³C NMR to determine Schiff base formation

Samples t = 0-8 hours, incubate at 23 °C

Samples t = 18-72 hours, incubate at 37 °C

5 mg (1 equiv, 0.02 mmol) of KBU2046 was dissolved in 0.35 mL of *d*-DMSO in an eppendorf tube. In a separate eppendorf tube, 1.9 mg (1 equiv, 0.02 mmol) of L-alanine was dissolved in 0.35 mL of D₂O. The L-alanine solution was added as a single portion to the solution of KBU2046, and the mixture was vortexed in (5 x 10 seconds) to ensure complete dissolution of all materials (combined solution was homogenous). KBU2046/L-alanine solution was added to a NMR tube and analyzed by ¹H and ¹³C NMR to determine if there was Schiff base formation. Reaction was incubated for 8 hours at 23 °C, where samples were taken at 1 hr, 4 hr and 8 hr timepoints, followed by an additional 64 hours at 37 °C, where samples were taken at 18 hr, 24 hr, 48 hr and 72 hr (relative to t = 0).

¹H NMR: L-alanine (1:1, *d*-DMSO: D₂O)

¹H NMR (500 MHz, DMSO-*d*₆) δ 3.52 (qd, *J* = 7.3, 6.6, 2.5 Hz, 1H), 1.28 (d, *J* = 7.2 Hz, 3H).

¹³C NMR: L-alanine (1:1, *d*-DMSO: D₂O)

¹³C NMR (125 MHz, DMSO) δ 176.12, 51.74, 17.74

¹H NMR: KBU2046 (1:1, *d*-DMSO: D₂O)

¹H NMR (500 MHz, DMSO-*d*₆) δ 7.75 (dd, $J = 7.9, 1.8$ Hz, 1H), 7.55 (ddd, $J = 8.7, 7.2, 1.8$ Hz, 1H), 7.22 (ddd, $J = 8.7, 5.5, 2.7$ Hz, 2H), 7.13 – 6.99 (m, 4H), 4.67 – 4.55 (m, 2H), 4.07 (dd, $J = 8.5, 5.1$ Hz, 1H).

¹³C NMR: KBU2046 (1:1, *d*-DMSO: D₂O)

¹³C NMR (125 MHz, DMSO) δ 194.39, 163.68, 162.37, 161.74, 138.10, 132.48, 128.15, 123.03, 121.32, 119.07, 116.74, 116.57, 71.78, 51.49.

¹H NMR: KBU2046: alanine-add, 5 min (1:1, *d*-DMSO: D₂O)

¹H NMR (500 MHz, DMSO-*d*₆) δ 7.76 (dd, *J* = 7.9, 1.8 Hz, 1H), 7.56 (ddd, *J* = 8.7, 7.2, 1.8 Hz, 1H), 7.28 – 7.20 (m, 2H), 7.14 – 7.03 (m, 3H), 7.06 – 7.01 (m, 1H), 4.68 – 4.56 (m, 2H), 4.28 (d, *J* = 4.5 Hz, 1H), 4.08 (dd, *J* = 8.6, 5.1 Hz, 1H), 3.39 (q, *J* = 7.2 Hz, 1H), 1.27 (d, *J* = 7.2 Hz, 2H).

¹³C NMR: KBU2046: alanine-add, 5 min (1:1, *d*-DMSO: D₂O)

¹³C NMR (125 MHz, DMSO) δ 194.39, 174.14, 163.68, 162.36, 161.74, 138.10, 132.48, 131.55, 128.15, 123.03, 121.32, 119.07, 116.74, 116.57, 71.78, 51.49, 50.99, 17.57.

¹H NMR: KBU2046: alanine-add, 1 hr (1:1, *d*-DMSO: D₂O)

¹H NMR (500 MHz, DMSO-*d*₆) δ 7.76 (dd, *J* = 7.9, 1.8 Hz, 1H), 7.56 (ddd, *J* = 8.7, 7.2, 1.8 Hz, 1H), 7.28 – 7.20 (m, 2H), 7.14 – 7.03 (m, 3H), 7.06 – 7.01 (m, 1H), 4.68 – 4.56 (m, 2H), 4.28 (d, *J* = 4.5 Hz, 1H), 4.08 (dd, *J* = 8.6, 5.1 Hz, 1H), 3.39 (q, *J* = 7.2 Hz, 1H), 1.27 (d, *J* = 7.2 Hz, 2H).

¹³C NMR: KBU2046: alanine-add, 1 hr (1:1, *d*-DMSO: D₂O)

¹³C NMR (125 MHz, DMSO) δ 194.39, 174.14, 163.68, 162.36, 161.74, 138.10, 132.48, 131.55, 128.15, 123.03, 121.32, 119.07, 116.74, 116.57, 71.78, 51.49, 50.99, 17.57.

¹H NMR: KBU2046: alanine-add, 4 hr (1:1, *d*-DMSO: D₂O)

¹H NMR (500 MHz, DMSO-*d*₆) δ 7.76 (dd, $J = 7.9, 1.7$ Hz, 1H), 7.56 (ddd, $J = 8.6, 7.2, 1.8$ Hz, 1H), 7.28 – 7.20 (m, 2H), 7.14 – 7.01 (m, 4H), 4.68 – 4.56 (m, 2H), 4.31 – 4.26 (m, 1H), 4.08 (dd, $J = 8.6, 5.1$ Hz, 1H), 3.39 (q, $J = 7.2$ Hz, 1H), 1.27 (d, $J = 7.2$ Hz, 3H).

¹³C NMR: KBU2046: alanine-add, 4 hr (1:1, *d*-DMSO: D₂O)

¹³C NMR (125 MHz, DMSO) δ 194.36, 174.12, 163.66, 162.34, 161.72, 138.08, 132.44, 131.53, 128.13, 123.00, 121.30, 119.05, 116.72, 116.55, 71.76, 51.47, 50.97, 17.55.

¹H NMR: KBU2046: alanine-add, 8 hr (1:1, *d*-DMSO: D₂O)

¹H NMR (500 MHz, DMSO-*d*₆) δ 7.76 (dd, *J* = 7.9, 1.8 Hz, 1H), 7.56 (ddd, *J* = 8.7, 7.3, 1.8 Hz, 1H), 7.28 – 7.20 (m, 2H), 7.14 – 7.01 (m, 4H), 4.68 – 4.56 (m, 2H), 4.08 (dd, *J* = 8.6, 5.1 Hz, 1H), 3.39 (q, *J* = 7.2 Hz, 1H), 1.27 (d, *J* = 7.2 Hz, 3H).

¹³C NMR: KBU2046: alanine-add, 8 hr (1:1, *d*-DMSO: D₂O)

¹³C NMR (125 MHz, DMSO) δ 194.37, 174.12, 163.66, 162.35, 161.72, 138.08, 132.44, 131.53, 128.13, 123.01, 121.30, 119.05, 116.72, 116.55, 71.76, 51.47, 50.97, 17.55.

¹H NMR: KBU2046: alanine-add, 18 hr (1:1, *d*-DMSO: D₂O)

¹H NMR (500 MHz, DMSO-*d*₆) δ 7.76 (dd, *J* = 7.8, 1.8 Hz, 1H), 7.56 (ddd, *J* = 8.7, 7.2, 1.8 Hz, 1H), 7.24 (ddd, *J* = 8.8, 5.6, 2.7 Hz, 2H), 7.08 (dt, *J* = 13.4, 8.3 Hz, 3H), 7.03 (d, *J* = 8.4 Hz, 1H), 4.68 – 4.56 (m, 2H), 4.32 – 4.27 (m, 1H), 4.08 (dd, *J* = 8.6, 5.1 Hz, 1H), 3.39 (q, *J* = 7.2 Hz, 1H), 1.27 (d, *J* = 7.2 Hz, 3H).

¹³C NMR: KBU2046: alanine-add, 18 hr (1:1, *d*-DMSO: D₂O)

¹³C NMR (125 MHz, DMSO) δ 194.37, 174.12, 163.66, 162.35, 161.72, 138.08, 132.44, 131.53, 128.13, 123.01, 121.30, 119.05, 116.72, 116.55, 71.76, 51.47, 50.97, 17.55.

¹H NMR: KBU2046: alanine-add, 24 hr (1:1, *d*-DMSO: D₂O)

¹H NMR (500 MHz, DMSO-*d*₆) δ 7.76 (dd, *J* = 7.9, 1.8 Hz, 1H), 7.56 (ddd, *J* = 8.7, 7.1, 1.8 Hz, 1H), 7.24 (ddd, *J* = 8.5, 5.4, 2.5 Hz, 2H), 7.14 – 7.03 (m, 3H), 7.03 (d, *J* = 8.4 Hz, 1H), 4.68 – 4.56 (m, 2H), 4.29 (s, 1H), 4.22 – 3.94 (m, 0H), 4.08 (dd, *J* = 8.5, 5.1 Hz, 1H), 3.39 (qd, *J* = 7.2, 2.4 Hz, 1H), 1.26 (dd, *J* = 7.2, 2.4 Hz, 3H).

¹³C NMR: KBU2046: alanine-add, 24 hr (1:1, *d*-DMSO: D₂O)

¹³C NMR (125 MHz, DMSO) δ 194.37, 174.11, 163.66, 162.35, 161.72, 138.08, 132.43, 131.53, 128.13, 121.30, 119.05, 116.72, 116.55, 71.76, 71.71, 51.47, 50.97, 17.55.

¹H NMR: KBU2046: alanine-add, 48 hr (1:1, *d*-DMSO: D₂O)

¹H NMR (500 MHz, DMSO-*d*₆) δ 7.76 (dd, *J* = 7.9, 1.8 Hz, 1H), 7.56 (ddd, *J* = 8.7, 7.2, 1.8 Hz, 2H), 7.24 (qd, *J* = 7.9, 6.7, 3.3 Hz, 4H), 7.14 – 7.00 (m, 6H), 4.68 – 4.57 (m, 2H), 4.09 (dd, *J* = 8.5, 5.1 Hz, 1H), 3.39 (q, *J* = 7.2 Hz, 1H), 2.47 (d, *J* = 14.1 Hz, 3H), 1.26 (t, *J* = 6.9 Hz, 5H).

¹³C NMR: KBU2046: alanine-add, 48 hr (1:1, *d*-DMSO: D₂O)

¹³C NMR (125 MHz, DMSO) δ 194.39, 174.12, 162.35, 138.09, 132.45, 131.56, 131.50, 128.13, 123.01, 121.30, 119.05, 116.72, 116.56, 71.76, 71.72, 51.47, 50.97, 17.55.

¹H NMR: KBU2046: alanine-add, 72 hr (1:1, *d*-DMSO: D₂O)

¹H NMR (500 MHz, DMSO-*d*₆) δ 7.76 (dd, *J* = 7.9, 1.8 Hz, 1H), 7.56 (ddd, *J* = 8.7, 7.2, 1.8 Hz, 1H), 7.24 (ddd, *J* = 8.6, 5.4, 2.6 Hz, 2H), 7.14 – 7.01 (m, 4H), 4.68 – 4.56 (m, 2H), 4.08 (dd, *J* = 8.5, 5.1 Hz, 1H), 3.39 (q, *J* = 7.2 Hz, 1H), 1.27 (d, *J* = 7.2 Hz, 3H).

¹³C NMR: KBU2046: alanine-add, 72 hr (1:1, *d*-DMSO: D₂O)

¹³C NMR (125 MHz, DMSO) δ 194.39, 174.12, 162.35, 138.09, 132.45, 131.56, 131.50, 128.13, 123.01, 121.30, 119.05, 116.72, 116.56, 71.76, 71.72, 51.47, 50.97, 17.55.

REVIEWERS' COMMENTS:

Reviewer #4 (Remarks to the Author):

The manuscript entitled "Precision Therapeutic Targeting of Human Cancer Cell Motility" continues to seriously concern me and I am not comfortable with its publication in Nature Communications. After several iterations of the submission, the authors have improved their argument and refined the claims. However, I disagree with the authors claims that their work has produced a small molecule with highly selective biological activity without off-target toxicity.

Response to Reviewers' comments:

Please find below our responses to reviewer #4's review. Associated changes in the manuscript are underlined.

Reviewer #4:

The manuscript entitled "Precision Therapeutic Targeting of Human Cancer Cell Motility" continues to seriously concern me and I am not comfortable with its publication in Nature Communications. After several iterations of the submission, the authors have improved their argument and refined the claims. However, I disagree with the authors claims that their work has produced a small molecule with highly selective biological activity without off-target toxicity.

Response. We respectfully do not agree with this reviewer's assessment. We stand by our prior response. In very broad terms, the investigations associated with this manuscript are comprehensive in both depth and breadth, our prior response contained new experimental data demonstrating lack of Schiff-base base formation, further bolstering the selective biological activity of KBU2046, and assessment by an additional biochemistry expert supports publication.